# Coarse-to-Fine Contrastive Learning in Image-Text-Graph Space for Improved Vision-Language Compositionality

**Harman Singh[1†], Pengchuan Zhang[1], Qifan Wang[1],**
**Mengjiao Wang[1], Wenhan Xiong[1], Jingfei Du[1†], Yu Chen[2†]**
[1]Meta AI    [2]Anytime.AI

harmansingh.iitd@gmail.com
{pengchuanzhang, wqfcr, mengjiaow, xwhan, jingfeidu}@meta.com
ychen@anytime-ai.com

## Abstract

Contrastively trained vision-language models have achieved remarkable progress in vision and language representation learning. However, recent research has highlighted severe limitations of these models in their ability to perform compositional reasoning over objects, attributes, and relations. Scene graphs have emerged as an effective way to understand images compositionally. These are graph-structured semantic representations of images that contain objects, their attributes, and relations with other objects in a scene. In this work, we consider the scene graph parsed from text as a proxy for the image scene graph and propose a graph decomposition and augmentation framework along with a coarse-to-fine contrastive learning objective between images and text that aligns sentences of various complexities to the same image. We also introduce novel negative mining techniques in the scene graph space for improving attribute binding and relation understanding. Through extensive experiments, we demonstrate the effectiveness of our approach that significantly improves attribute binding, relation understanding, systematic generalization, and productivity on multiple recently proposed benchmarks (For example, improvements up to $18\%$ for systematic generalization, $16.5\%$ for relation understanding over a strong baseline), while achieving similar or better performance than CLIP on various general multimodal tasks.

## 1 Introduction

Recent progress in contrastive learning using large-scale image-text data for joint image-text representation learning has led to Vision-Language models (VLMs) like CLIP (Radford et al., 2021) and ALIGN (Jia et al., 2021) that show remarkable zero-shot classification and retrieval capabilities. However, recent works have shown that these models struggle at compositional reasoning (Yuksekgonul

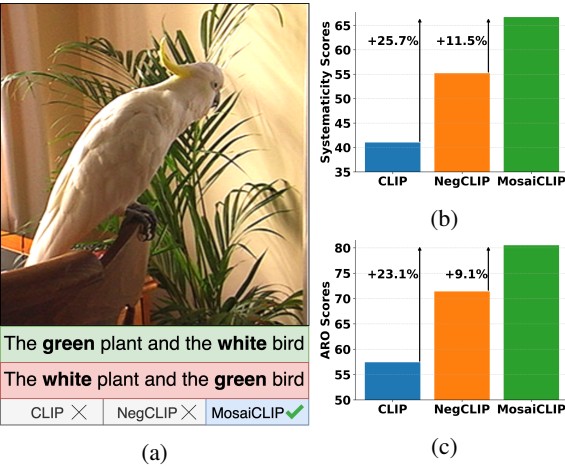

Figure 1: **(Left)** a) A typical example from the ARO benchmark for testing attribute understanding of VLMs. VLMs struggle with matching the image to the correct caption (in green). **(Right)** Average scores of MosaiCLIP (our method) compared with NegCLIP and CLIP on prominent compositionality benchmarks for measuring b) Systematic Generalization c) Attribute, Relation, and Word Order understanding.

et al., 2022; Thrush et al., 2022; Ma et al., 2022). In particular, they struggle with binding correct attributes to the correct objects, understanding relations between objects, generalizing systematically to unseen combinations of concepts and to larger and more complex sentences.

Some works have made progress on this problem. Yuksekgonul et al. (2022) show that hard negative mining of images and text during fine-tuning is a promising first step to improving compositionality. However, performance gains are highly dependent on how clean the training data is, and generalizing to unseen combinations of concepts remains a challenge. Doveh et al. (2023) use LLMs for hard negative mining and Cascante-Bonilla et al. (2023) explore using synthetic datasets to improve compositional understanding in VLMs. Synthetic datasets lead to a domain gap compared to natural datasets. We aim to develop a general-purpose approach for improving compositionality of all such

---

[†] Work done while at Meta.

contrastively trained VLMs.

In this paper, we consider a scene graph representation of the image and text. We observe that multiple sub-graphs of the text scene graph with different semantic complexities can be matched with the same image. Performing this matching improves fine-grained and hierarchical understanding of text and thereby, of images. We achieve this by developing a scene graph-based text decomposition strategy that creates a scene graph for any given text, decomposing it into sub-graphs, and matching an image to multiple sentences derived from these sub-graphs (See Fig. 2 for an overview). Each sub-graph represents a distinct part of the image, aligning well with CLIP's original image-text matching objective. Focused on improving attribute binding and relation understanding, we develop novel hard negative graph creation strategies which helps VL contrastive learning. We provide a novel Image-to-Multi-Text contrastive loss for matching individual images to multiple sentences. Our approach of matching texts of different complexity (from coarse-grained to fine-grained) to the image leads to fine-grained and hierarchical text understanding. Our resulting model is MosaiCLIP.

Our approach leads to significant improvements across compositionality benchmarks. For example, Figure 1 b) and c) shows that MosaiCLIP improves performance by 11.5% and 9.1% on CREPE and ARO dataset over a strong baseline and by > 20% over CLIP. **Our contributions encompass:**

- A novel graph-based text decomposition and augmentation framework and a coarse-to-fine contrastive learning objective for matching images to text sub-graphs of varying complexity.

- Hard-negative mining techniques using graph transformations of the text scene graphs, that are seamlessly coupled with our text decomposition strategy, and applied over any text.

- A thorough *analysis* for understanding why MosaiCLIP improves vision-language compositionality, disentangling the effect of image and text encoders and providing a novel tree-score based analysis showing that MosaiCLIP exhibits improved hierarchical text understanding.

- Extensive experiments over three model architectures, two pre-training datasets, three fine-tuning datasets and test over four compositionality benchmarks (11 datasets) to prove the efficacy of MosaiCLIP for improving compositionality.

## 2 Related Work

**Contrastive Vision-Language Pre-training:** Large-scale contrastive learning for Vision and Language is utilized to create models like CLIP (Radford et al., 2021) and ALIGN (Jia et al., 2021). These models showcase impressive performance on a variety of tasks, including image classification, text and image retrieval, image captioning (Mokady et al., 2021), object detection (Zhong et al., 2022; Li et al., 2022c) etc.

**Visio-Linguistic Compositionality:** Various studies have introduced benchmarks for assessing the compositional reasoning abilities of vision-language foundation models (VLMs). For instance, Winoground (Thrush et al., 2022) is a handpicked collection of 400 test cases, each comprising two images and two sentences. Sentences have the same word content and differ in word-order. Diwan et al. (2022) show that the Winoground dataset tests additional challenges along with compositionality, including handling ambiguous image-text pairs and unusual examples. Yuksekgonul et al. (2022) proposed the ARO benchmark for probing VLMs ability to understand Attribute, Relations, and Word-Order. Ma et al. (2022) proposed CREPE for measuring two aspects of compositionality: systematic generalization and productivity. All benchmarks suggest that contrastively trained VLMs have severe difficulty in compositional reasoning. As a remedy, NegCLIP (Yuksekgonul et al., 2022) and Teaching SVLC (Doveh et al., 2023) create targeted rule-based and LLM-guided hard negative sentences, SyViC (Cascante-Bonilla et al., 2023) fine-tunes CLIP with million scale synthetic images-text pairs, for improving relational and attribute understanding. We observe that previous methods are either highly dependent on how clean the training data is, use expensive LLM's for data augmentation or use synthetic datasets that require special solutions to resolve the synthetic-to-real domain gap. We hence develop a coarse-to-fine contrastive learning framework that matches images with texts of multiple complexities, which serves as a general-purpose solution to improve fine-grained and hierarchical text understanding,

thereby improving compositionality.

**Scene Graphs** are structured representations of visual scenes, consisting of objects, their attributes, and relationships between objects. Scene graphs are beneficial for a range of tasks including image retrieval (Wu et al., 2019; Johnson et al., 2015), image captioning (Yang et al., 2019), and image generation (Johnson et al., 2018) among others.

# 3 Methodology

## 3.1 Overview

Here we present the key high-level ideas of our approach. We first present a graph-centric view of the standard image-text matching objective in CLIP, which serves as a motivation to develop our approach (Sec. 3.2). We create scene graphs derived from the text, decompose them into multiple sub-graphs (Sec. 3.3) and apply augmentations on these sub-graphs to create negative sub-graphs (Sec. 3.4) which are used as hard negatives in a batch. Sec. 3.5 formally defines the Image-to-Multi-Text and Text-to-Image losses used for a batch of V-L inputs which is key for learning from multiple positive and negative texts derived from sub-graphs. Matching images with coarse-to-fine sub-graphs results in improved fine-grained and hierarchical understanding of text. Sec. 3.6 provides a two-stage curriculum learning strategy for improved fine-tuning performance.

## 3.2 Image-Text-Graph Alignment

Our approach builds on the idea that the standard image-text contrastive learning in CLIP can be viewed as a matching between an image scene graph and its sub-graph. Formally, given an image-text pair $(I, T)$, the image can be viewed by its scene graph, $\mathcal{G}_I = (\mathcal{V}_I, \mathcal{E}_I)$. The text scene graph is given by $\mathcal{G}_T = (\mathcal{V}_T, \mathcal{E}_T)$. Then $\mathcal{G}_T \subset \mathcal{G}_I$. According to this assumption, during contrastive learning in CLIP, we implicitly bring the representation of the image scene graph close to *one* of its sub-graph (the text scene graph). Now, let $S_{\mathcal{G}} = \{g | g \subset \mathcal{G}\}$ represent the set of sub-graphs of a graph $\mathcal{G}$. According to the assumption above, $g \in S_{\mathcal{G}_T} \Rightarrow g \in S_{\mathcal{G}_I}$. Hence $\forall g \in S_{\mathcal{G}_T}, (g, \mathcal{G}_I)$ becomes a correct matching pair during contrastive learning. We match multiple sub-graphs of the text scene graph to the same image, while also including hard negative sub-graphs in the batch. Matching between graphs is an implicit concept,

and all graphs are first converted to text via templates, converted to embeddings using transformer-based (text) encoders, and matched to image embeddings.

## 3.3 Scene Graph Guided Text Decomposition

Scene graphs are succinct representations of images. However, an image scene graph generator used for generating a scene graph for any given input image is expensive to train since it requires supervised scene graph annotations for training (Li et al., 2017; Xu et al., 2017; Zhang et al., 2019), and also leads to issues like low coverage or biased generations against the long tail nature of objects and relationship annotations. We instead use the text scene graph created using an off-the-shelf text scene graph parser[1] (Wu et al., 2019). This serves as a proxy for the scene graph of (part of) the image and is assumed to be a sub-graph of the image scene graph, as also depicted by Figure 2.

Let the text scene graph obtained be $G_T = (V_T, E_T)$, where $V_T$ represent the nodes of the graph, which are either objects or their attributes. $E_T$ are the edges of the graph that represent relations between objects. See Fig. 2 for an example of a text scene graph. As shown in the figure, we decompose this scene graph into multiple *positive* sub-graphs $P_g = \{g_1, g_2, g_3, \cdots, g_k\}$, $k \leq M$, where $M$ is the max number of decomposed sub-graphs and is a hyperparameter. Each sub-graph is a representation of a part of the image. We then convert sub-graphs to sentences so that they can be easily processed by transformer-based (text) encoders commonly used to train CLIP. For this, we use a simple template-based approach. For e.g., we create templates of the form "$\{N_1\} \{R\} \{N_2\}$" if we need to convert a graph having two nodes $(N_1, N_2)$ and a relation $R$, into a sentence format. Corresponding to each sub-graph, we obtain one positive text for the image, creating a positive text set $P_t = \{t_1, t_2, t_3, \cdots, t_k\}$.

## 3.4 Negative Sub-Graph Creation

Corresponding to sub-graphs in $P_g$, we create negative sub-graphs $N_g = \{{}^n g_1, {}^n g_2, {}^n g_3, \cdots\}$. Sub-graphs in $N_g$ are a minimally perturbed versions of the positive sub-graphs in $P_g$. Similar to positive sub-graphs, we convert sub-graphs in $N_g$ to text using the same template-based approach, and obtain $N_t = \{{}^n t_1, {}^n t_2, {}^n t_3, \cdots\}$. Texts in $N_t$ serve

---

[1] https://github.com/vacancy/SceneGraphParser

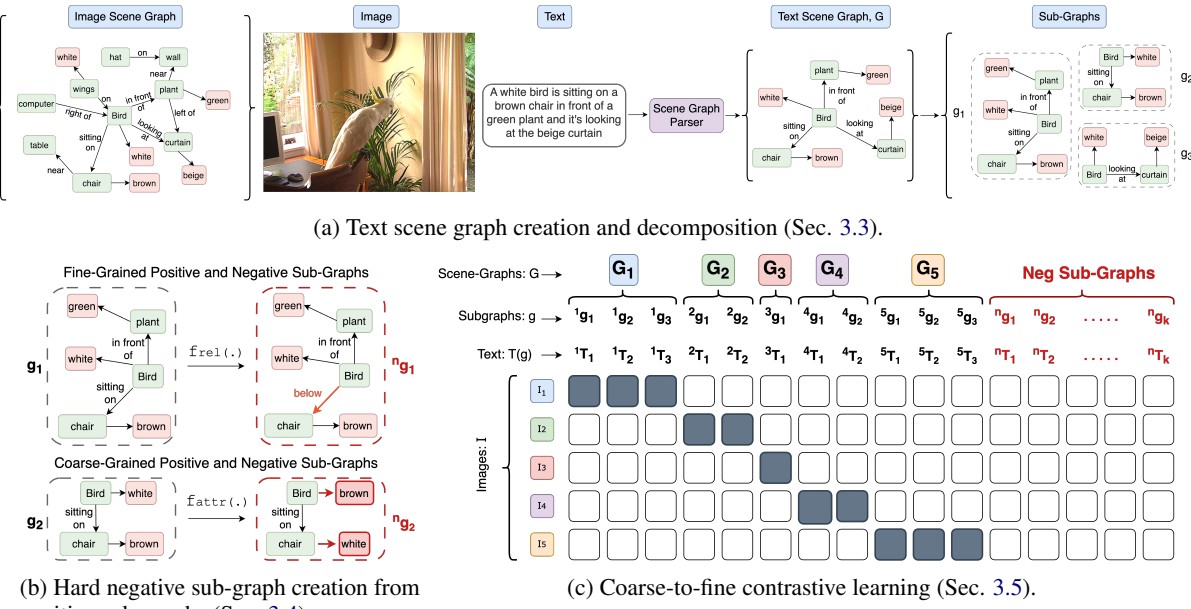

(a) Text scene graph creation and decomposition (Sec. 3.3).

(b) Hard negative sub-graph creation from positive sub-graphs (Sec. 3.4).

(c) Coarse-to-fine contrastive learning (Sec. 3.5).

Figure 2: Overview of our approach. **a)** Depiction of the scene graph of an image (hypothetical) and a scene graph parsed from text. The text scene graph is a sub-graph of the image scene graph. The text scene graph is decomposed into sub-graphs from which **b)** minimally perturbed hard-negative sub-graphs are created. **c)** The Ground truth similarity matrix used for a batch of data during contrastive learning. Solid boxes represent a match between the image and the corresponding text. Different from CLIP, each image can be matched to multiple texts in our method.

as hard negative texts in a given batch, see Fig. 2. We focus on creating negative sub-graphs that improve the attribute binding and relation understanding capabilities of the model, for which we use the following strategies for negative graph creation: We first consider an external set of objects ($\mathcal{N}$), attributes ($\mathcal{A}$), and relations ($\mathcal{R}$).
**1) Node Swapping and Replacement**: We *swap* nodes in sub-graphs, these can be swaps of nodes which are attributes or objects. We also *replace* nodes with external nodes from $\mathcal{N}$, $\mathcal{A}$ based on their type. **2) Edge Replacement**: We *replace* edges with randomly sampled edges from the external relations set, $\mathcal{R}$. **3) Connecting Sub-graphs**: Here we *join* two sub-graphs. For this, we use one sub-graph from $P_g$, and another random graph created using nodes and edges sampled from external sets $\mathcal{N}, \mathcal{A}, \mathcal{R}$. This creates an overall hard negative graph. Sub-graphs are joined by simply joining nodes from both graphs through a randomly sampled edge from $\mathcal{R}$. These strategies result in minimally perturbed hard negative sub-graphs for improving attribute and relation understanding. We define multiple graph transformations $\{f_g : \mathbb{G} \longrightarrow P(\mathbb{G})\} - f_{rel}, f_{attr}, f_{obj}$ using the above techniques and create hard negative sub-graphs. See Appendix Sec. B for more details regarding negative sub-graph creation.

## 3.5 Coarse-to-Fine Contrastive Learning in Image-Text-Graph Space

Given an image-text batch during training $\mathcal{B} = \{(\boldsymbol{x}_i, \boldsymbol{t}_i)\}_{i=1}^{n}$, consider separately the batch of images $\mathcal{B}_I = \{\boldsymbol{x}_i\}_{i=1}^{n}$ and a batch of texts $\mathcal{B}_T = \{\boldsymbol{t}_i\}_{i=1}^{n}$. The sentences in the text batch are first converted to scene graphs to obtain a batch of scene graphs $\mathcal{B}_G = \{\mathcal{G}_i\}_{i=1}^{n}$, followed by decomposition to sub-graphs to obtain the positive sub-graph batch $\mathcal{B}_g^{pos} = \{\boldsymbol{g}_i\}_{i=1}^{m}$, $m > n$. $r$ negative sub-graphs are sampled and added to the batch to obtain $\mathcal{B}_g = \{\boldsymbol{g}_i\}_{i=1}^{m+r}$. We convert these sub-graphs to text to obtain the final text batch $\mathcal{B}_t = \{\boldsymbol{t}_i^g\}_{i=1}^{m+r}$.

Consider an image encoder model $f_\theta$ parameterized by $\theta$, a text encoder $f_\phi$ parameterized by $\phi$. For any image $\boldsymbol{x}$, text $\boldsymbol{t}$, $\tilde{\boldsymbol{u}} = f_\theta(\boldsymbol{x})$ is the unnormalized image feature, and $\tilde{\boldsymbol{v}} = f_\phi(\boldsymbol{t})$ is the unnormalized text feature. As common practice, the features are normalized to obtain $\boldsymbol{u} = \tilde{\boldsymbol{u}}/\|\tilde{\boldsymbol{u}}\|$ and $\boldsymbol{v} = \tilde{\boldsymbol{v}}/\|\tilde{\boldsymbol{v}}\|$. The **Image-to-Multi-Text** contrastive loss is given by:

$$\mathcal{L}_{i2t}^{\text{MC}} = -\sum_{i=1}^{|\mathcal{B}_I|} \frac{1}{|\mathcal{P}(i)|} \sum_{k \in \mathcal{P}(i)} \log \frac{\exp(\tau \boldsymbol{u}_i^T \boldsymbol{v}_k)}{\sum_{j=1}^{|\mathcal{B}_t|} \exp(\tau \boldsymbol{u}_i^T \boldsymbol{v}_j)}$$

where $\mathcal{P}(i) = \{k | k \in [1, |\mathcal{B}_t^{pos}|], \boldsymbol{g}_k \subseteq \mathcal{G}_i\}$. The **Text-to-Image** contrastive loss is only calcu-

lated for the positive texts. It is given by:

$$\mathcal{L}_{t2i}^{\text{MC}} = - \sum_{j=1}^{|\mathcal{B}_t^{pos}|} \log \frac{\exp(\tau \boldsymbol{u}_{p(j)}^T \boldsymbol{v}_j)}{\sum_{i=1}^{|\mathcal{B}_I|} \exp(\tau \boldsymbol{u}_i^T \boldsymbol{v}_j)}$$

where $\boldsymbol{g}_{p(j)} \subseteq \mathcal{G}_j$. $\mathcal{B}_t = [\mathcal{B}_t^{pos}; \mathcal{B}_t^{neg}]$, in which $\mathcal{B}_t^{pos}, \mathcal{B}_t^{neg}$ represent the texts in $\mathcal{B}_t$, obtained from positive and negative sub-graphs respectively. The overall loss is $\mathcal{L}_{\text{MosaiCLIP}} = (\mathcal{L}_{t2i}^{\text{MC}} + \mathcal{L}_{i2t}^{\text{MC}})/2$.

### 3.6 Curriculum and Robust Fine-tuning

For fine-tuning experiments, we develop a two-stage curriculum learning strategy motivated by recent work (Goyal et al., 2022; Wortsman et al., 2022; Kumar et al., 2022) that show how fine-tuning can distort pre-trained features and closely mimicking the contrastive pre-training objective while fine-tuning CLIP can help mitigate this problem (Goyal et al., 2022). However, our coarse-to-fine contrastive learning objective naturally deviates from pre-training in two ways. *a)* Existence of hard negative texts in the batch, and *b)* Having multiple positive and negative texts for an image. This can lead to a *gap* in pre-training vs fine-tuning objective, and a lower than optimal performance after fine-tuning. To solve this, our two-stage curriculum learning strategy first fine-tunes the model while sampling (at max) a single positive and negative sub-graph per image, followed by fine-tuning it with multiple positive and negative sub-graphs. The hardness of data in this curriculum learning setup is defined by the amount of difference the fine-tuning setup has as compared to the pre-training setup. According to this intuition, it is easier for the model to first learn to handle hard negatives in a batch and then learn to handle multiple positive and hard negative sentences at once. We see consistent improvements using this strategy compared to a direct one-step fine-tuning, which we term as MosaiCLIP$_{\text{NoCurric}}$ in our ablations. For better performance on non-compositonal tasks, we use the robust fine-tuning approach (Wortsman et al., 2022) of weight space ensembling of the vision encoder, before and after fine-tuning. This model is called MosaiCLIP$_{\text{WiSE-FT}}$

## 4 Experiments

**Evaluation Datasets:** We test MosaiCLIP and baselines on large scale benchmarks that require compositional reasoning: CREPE-Systematicity (Ma et al., 2022) measures systematic generalization, ARO (Yuksekgonul et al., 2022) measures

attribute, relation and word-order understanding, SVO (Hendricks and Nematzadeh, 2021) measures verb (relation) understanding, VL-Checklist (Zhao et al., 2022) measures relation, attribute and object understanding. We use CREPE-Productivity (Ma et al., 2022) for measuring model's ability to productively generalize to more complex and long sentences. Methods for improving compositionality should be tested on general downstream tasks used to evaluate the quality of learned representations of language and vision. For this, we utilize the popular ELEVATER benchmark (Li et al., 2022a) consisting of 20 datasets and ImageNet (Deng et al., 2009) following prior work (Doveh et al., 2023).

**Baselines:** We compare with all recent techniques used for improving compositionality of CLIP style models including NegCLIP (Yuksekgonul et al., 2022), Teaching SVLC (Doveh et al., 2023) and Syn-CLIP (Cascante-Bonilla et al., 2023) along with CLIP (Radford et al., 2021) as well as CLIP-FT (fine-tuned) on datasets we use. See Appendix Sec. F for more details.

**Training and Evaluation Details:**
Fine-tuning: NegCLIP (Yuksekgonul et al., 2022) was developed by fine-tuning CLIP on the COCO dataset (Lin et al., 2014), however, COCO images might overlap with benchmarks like CREPE and ARO which may lead to confounding of results. Hence we consider 2 additional similar sized fine-tuning datasets randomly sampled from CC-12M (Sharma et al., 2018; Changpinyo et al., 2021) and YFCC-15M (Thomee et al., 2016) and call them CC-FT, YFCC-FT. We also use CC3M (Sharma et al., 2018) for comparing with recent baselines. We fine-tune the commonly used OpenAI CLIP-ViT-B32 model and report results on all datasets, except for CREPE dataset which tests the *systematic generalization* for which we used OpenCLIP (Ilharco et al., 2021) models pre-trained on {CC-12M, YFCC-15M}, fine-tune them on {CC-FT, YFCC-FT}, and report results on {CC-12M,YFCC-15M} splits of CREPE. See Appendix E.3 for more information on evaluation datasets.

Pre-training: We pre-train MosaiCLIP, NegCLIP and CLIP on two prominent large-scale pre-training datasets, CC-12M and YFCC-15M, and use two different backbones (ResNet-50 and Swin-Tiny) following prior work (Yang et al., 2022) and report zero-shot performance on all test datasets. See Appendix H.1 for hyperparameters details.

| FineTun. data → | COCO | | | | | CC-FT | | | | | | | YFCC-FT | | | | | | | Meta |
|---|---|---|---|---|---|---|---|---|---|---|---|---|---|---|---|---|---|---|---|---|
| Benchmark → | **ARO** | | | **VLC** | **SVO** | **ARO** | | | **CREPE** | | **VLC** | **SVO** | **ARO** | | | **CREPE** | | **VLC** | **SVO** | |
| Method ↓ | Rel. | Attr. | Ord. | Avg. | Avg. | Rel. | Attr. | Ord. | CU | AU | Avg. | Avg. | Rel. | Attr. | Ord. | CU | AU | Avg. | Avg. | Avg. |
| Random | 50.0 | 50.0 | 20.0 | 50.0 | 50.00 | 50.0 | 50.0 | 20.0 | 14.3 | 20.0 | 50.0 | 50.00 | 50.0 | 50.0 | 20.0 | 14.3 | 20.0 | 50.0 | 50.00 | 38.35 |
| CLIP | 59.8 | 63.2 | 53.3 | 70.8 | 83.58 | 59.8 | 63.2 | 53.3 | 45.1 | 35.0 | 70.8 | 83.58 | 59.8 | 63.2 | 53.3 | 39.8 | 39.5 | 70.8 | 83.58 | 60.60 |
| CLIP-FT | 58.9 | 65.3 | 38.4 | 71.3 | 90.15 | 58.1 | 63.3 | 42.7 | 45.8 | 35.6 | 70.1 | 88.56 | 51.4 | 63.1 | 25.3 | 36.4 | 38.3 | 68.9 | 85.27 | 57.73 |
| NegCLIP | 81.7 | 72.7 | 85.7 | 75.6 | 90.20 | 71.5 | 65.4 | 84.5 | 53.1 | 37.5 | 72.4 | 88.36 | 57.8 | 63.1 | 52.1 | 38.8 | 39.0 | 70.4 | 83.90 | 67.57 |
| **MosaiCLIP** | **82.6** | **78.0** | **87.1** | **81.4** | **90.67** | **80.4** | **69.8** | **85.5** | **72.4** | **40.9** | **77.6** | **88.73** | **74.3** | **66.9** | **84.4** | **48.8** | **41.5** | **75.1** | **85.36** | **74.29** |

Table 1: Fine-tuning results on the ARO, CREPE - Systematicity, VL-Checklist (VLC) and SVO benchmark (total 10 datasets). Abbreviations – Rel.:= VG-Relation, Attr.:= VG-Attribution, Ord:=Average of ARO-Flickr and ARO-COCO Order results, CU: HN-Comp-Unseen, AU: HN-Atom-Unseen. See Sec. 4.1 for more details.

| FineTun. data → | CC3M | | | | | |
|---|---|---|---|---|---|---|
| Benchmark → | **VL-Checklist** | | | **ARO** | | |
| Method | Obj. | Attr. | Rel. | Rel. | Attr. | Ord. |
| CLIP | 81.6 | 67.6 | 63.1 | 59.9 | 63.6 | 53.3 |
| CLIP-FT | 79.0 | 64.7 | 54.3 | 41.7 | 59.3 | 25.2 |
| Syn-CLIP[1] | - | - | 70.4 | 69.4 | 71.4 | 66.9 | 65.1 |
| Teaching SVLC[2] | 85.0 | 72.0 | 69.0 | - - | - - | |
| **MosaiCLIP** | **86.4** | **73.7** | **71.9** | **83.7** | **78.0** | **79.4** |

[1](Cascante-Bonilla et al., 2023) [2](Doveh et al., 2023)

Table 2: MosaiCLIP vs contemporary works that use [†]synthetic data or [‡]LLM's. See Appx. D.1 for details.

## 4.1 Results

In this section we provide experimental results in both pre-training and fine-tuning settings to show the efficacy of our approach. These are as follows:

**Fine-tuning:** Main fine-tuning results are shown in Table 1 and 2, where we fine-tune CLIP models using our method and compare it to baselines. Notably, we see that the generalization performance on unseen compounds and atoms as measured by the CREPE dataset is up to 18% higher than NegCLIP. Additionally MosaiCLIP shows upto 16.5%, 5.3%, 32.3% of improvement over NegCLIP in understanding relations, attributes and word order respectively. MosaiCLIP also shows consistent improvements in the verb understanding task as measured by the SVO dataset. **Additional Comparisons**: We also compare with latest contemporary works in Table 2 and Appendix Sec. D.1. We find significant improvements (upto 14% on ARO) over models that use LLMs or synthetic data for making CLIP more compositonal.

**Pre-training:** Table 3 shows pre-training results over all benchmarks. CREPE results show

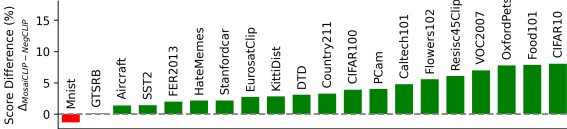

Figure 3: MosaiCLIP's average score difference with NegCLIP on 20 datasets from ELEVATER benchmark.

a significant gain in ability to systematically generalize to unseen combinations of concepts. Across pre-training settings, MosaiCLIP improves over NegCLIP by up to 42.5%, 4.9% when evaluated against HN-Comp (CU), HN-Atom (AU) hard negatives respectively. Significant improvements are observed in attribute and relation understanding, giving gains of up to 8.3%, 12.0% respectively across pretraining settings. We also note that order understanding of MosaiCLIP is worse than that of NegCLIP for the CC-12M pre-training dataset, while better than NegCLIP for the YFCC-15M dataset. Notably, there is a large variance in NegCLIP's performance across pre-training datasets as seen in Table 3, and it also performs poorly when the pre-training dataset has higher noise (e.g. YFCC-15M). MosaiCLIP is fairly consistent and more robust to the change in the pre-training dataset. In Appendix C.5 we find that MosaiCLIP can provide improvements over NegCLIP while using as low as **0.3x** of the total pre-training or fine-tuning data.

**Results on classification and retrieval**: On average, MosaiCLIP achieves +3.3%, +6.3% better performance on the ELEVATER classification benchmark compared to NegCLIP and CLIP while pre-training and maintains similar accuracy as CLIP while fine-tuning. We also try using our method along with the robust fine-tuning technique (WiSE-FT) so that performance degra-

| Arch. | Method ↓ | CC-12M | | | | | | | YFCC-15M | | | | | | | |
|---|---|---|---|---|---|---|---|---|---|---|---|---|---|---|---|---|
| | | **ARO** | | | **CREPE** | | **VLC** | **SVO** | **ARO** | | | **CREPE** | | **VLC** | **SVO** | Meta |
| | | Rel. | Attr. | Ord. | CU | AU | Avg. | Avg. | Rel. | Attr. | Ord. | CU | AU | Avg. | Avg. | Avg. |
| | Random | 50.0 | 50.0 | 20.0 | 14.3 | 20.0 | 50.0 | 50.00 | 50.0 | 50.0 | 20.0 | 14.3 | 20.0 | 50.0 | 50.00 | 36.33 |
| Swin-T | CLIP | 51.0 | 56.6 | 25.5 | 44.1 | 37.3 | 65.6 | 82.21 | 53.8 | 56.2 | 18.4 | 39.6 | 41.7 | 66.2 | 76.27 | 51.03 |
| | NegCLIP | 82.4 | 66.8 | **59.7** | 80.3 | 39.6 | 70.0 | 82.04 | 73.6 | 58.9 | 35.5 | 47.1 | 41.5 | 66.0 | 76.10 | 62.82 |
| | **MosaiCLIP** | **84.3** | **76.8** | 55.5 | **92.1** | **44.5** | **72.4** | **85.62** | **74.7** | **66.1** | **35.8** | **89.6** | **45.3** | **71.8** | **77.87** | **69.46** |
| RN-50 | CLIP | 52.9 | 59.7 | 22.6 | 42.9 | 36.7 | 66.2 | 82.13 | 57.8 | 55.1 | 18.3 | 38.9 | 38.9 | 64.8 | 75.60 | 50.90 |
| | NegCLIP | 80.5 | 66.5 | **60.5** | 82.0 | 41.4 | 69.5 | 82.03 | 68.0 | 58.5 | 37.1 | 67.2 | 41.5 | 66.1 | 75.18 | 64.00 |
| | **MosaiCLIP** | **82.0** | **78.5** | 55.4 | **92.6** | **44.4** | **72.6** | **83.86** | **76.3** | **68.9** | **38.2** | **90.2** | **45.0** | **72.3** | **77.42** | **69.83** |

Table 3: Pre-training results on all compositionality benchmarks (4 benchmarks, 10 datasets) over four expt. settings (two pre-training datasets, two backbones). See Table 1 for abbreviations and Sec. 4.1 for more details.

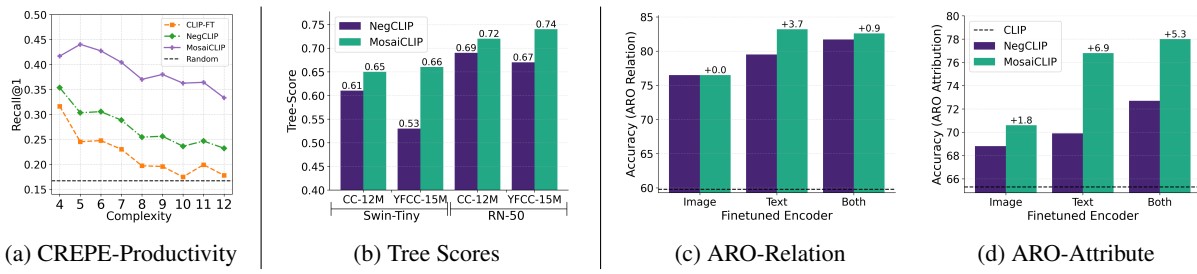

(a) CREPE-Productivity     (b) Tree Scores     (c) ARO-Relation     (d) ARO-Attribute

Figure 4: **a)** Results on CREPE-Productivity dataset **b)** Tree-score comparison of MosaiCLIP with NegCLIP: MosaiCLIP shows improved hierarchical understanding of language. **c) and d)** Selectively fine-tuning of image, text encoders and measure performance on different datasets. Also see similar results for SVO in Figure 10.

| Model | COCO | | Flickr30K | | AVG. |
|---|---|---|---|---|---|
| | I2T | T2I | I2T | T2I | |
| CLIP | 20.7 | 13.1 | 36.2 | 24.1 | 23.5 |
| NegCLIP | 20.1 | 12.9 | 38.6 | 23.3 | 23.7 |
| MosaiCLIP | **25.9** | **16.5** | **44.5** | **29.5** | **29.1** |

Table 4: Comparison of Recall@1 scores of MosaiCLIP with NegCLIP and CLIP. All models are pre-traind on YFCC-15M with swin-Tiny backbone

dation during fine-tuning is minimal, as shown in Appendix Table 9. See Fig. 3 for average results on ELEVATER over four training settings and Table 4 for results on retrieval benchmarks where we see a +5.4 point improvement over NegCLIP. We use the popular Karpathy splits having a 5K and 1K sized test set for obtaining the COCO and Flickr30k retrieval scores respectively. Hence MosaiCLIP's training strategy improves or maintains the quality of learned representations while improving compositonality. Figures 11-14 show detailed results on ELEVATER.

**Productivity**: As defined by Ma et al. (2022), a productive VL model can handle arbitrarily long and complex sentences and is an important

aspect of compositionality. Although we do not explicitly train our models for generalization to longer sentences, the improved hierarchical language understanding using our methods lead to an emergent behavior such that MosaiCLIP generalizes better than NegCLIP and CLIP to more complex sentences. We can see this effect in Fig. 4 a) and Appendix Fig. 8 and 9. We report the average of retrieval over swap and atom splits and find MosaiCLIP significantly improves over NegCLIP by upto 15% across different text complexities (4-12).

**Application to more advanced VLMs:** While our focus in this work has been on CLIP style, dual encoder models due to their various benefits, we believe our methods are model agnostic and aimed at improving contrastive learning through our coarse-to-fine learning framework and negative mining techniques. In this section we test our model on an advanced VLM, BLIP. We modified BLIP's original image-text contrastive learning objective and create two variants, one called BLIP+NegCLIP where we use NegCLIP style hard negatives and the other BLIP+MosaiCLIP which uses our methods of scene graph guided

text decomposition and negative sub-graph creation. We fine-tune BLIP model taken from the official BLIP repository and use the "BLIP w/ ViT-B and CapFilt-L model (pre-trained on 129M examples)" as our base model. Results for fine-tuning experiment using COCO dataset is shown in Table 5. We use the hyperparameters used by the official codebase (for the task of fine-tuning on COCO dataset for image-text retrieval). For each setting, we report performance of four models, namely BLIP (before fine-tuned version), BLIP-FT (vanilla fine-tuned version), BLIP+NegCLIP, BLIP+MosaiCLIP. The model are evaluated on the ARO dataset to measure attribute, relation and word-order understanding, using the evaluation scripts provided by the authors of the dataset (Yuksekgonul et al., 2022). We find that

| Model | Rel | Attr | Ord | Avg |
|---|---|---|---|---|
| BLIP | 53.5 | 91.0 | 53.5 | 66.0 |
| BLIP-FT | 58.9 | 88.4 | 58.9 | 68.7 |
| BLIP+NegCLIP | 63.6 | 90.7 | 63.6 | 72.6 |
| BLIP+MosaiCLIP | **69.9** | **91.1** | **69.9** | **77.0** |

Table 5: Comparison of BLIP (Li et al., 2022b) and fine-tuned version of BLIP with BLIP models that have integrated NegCLIP and MosaiCLIP methodology while training. Fine-tuning has been performed on COCO.

compared to vanilla fine-tuning, both NegCLIP and MosaiCLIP methodologies bring improvements to relation and word order understanding, while maintaining or improving performance on attribute understanding. The MosaiCLIP methodology significantly improves relational reasoning performance and word-order understanding compared to the NegCLIP methodology, up to 6.3%. Attribute understanding performance remains nearly the same as the baseline BLIP performance, with the MosaiCLIP methodology bringing in slight gains over NegCLIP's methodology. On average MosaiCLIP's methodology brings more improvements to BLIP than NegCLIP or vanilla fine-tuning.

### 4.2 Analysis

We provide a detailed analysis of our models and baselines, across different dimensions as follows:

**Disentangling MosaiCLIP improvements:** We quantify the relative importance of the vision and language side by freezing the language and vision encoder individually while fine-tuning all

models. See Fig. 4 c,d for the results. Notably, we find that **1)** Language encoder has significant scope for improvement over NegCLIP's language encoder, and MosaiCLIP is able to successfully exploit this potential and deliver an enhanced compositional understanding of language, which is evident by performance increase of $+3.7, +6.9\%$ over NegCLIP when only the language encoder is fine-tuned, as shown in Fig. 4 c,d. **2)** Improvements brought by MosaiCLIP over NegCLIP in the text encoder are always higher than improvements in the image encoder. This is evident from Fig. 4 c,d where the performance increase over NegCLIP when only the language encoder is fine-tuned is always higher as compared to when only the image encoder is fine-tuned; for example, $3.7\% > 0.0\%$, $6.9\% > 1.8\%$ for ARO-Relation, ARO-Attribution. **3)** MosaiCLIP brings significant improvements on the image encoder side (higher than NegCLIP) *without* using any image negative mining, unlike NegCLIP.

**MosaiCLIP improves hierarchical text understanding:** For further understanding MosaiCLIP's improved compositional understanding, we provide a novel analysis by considering the recently proposed Tree-Score (Murty et al., 2022) that measures the degree to which a transformer (text) encoder processes text in a hierarchical manner. We hypothesize that having tree-like hierarchical computation over language can be one leading factor for explaining the compositionality (or lack thereof) of CLIP-like models. Along with this, we have previously shown that the language encoder has the most prominent effect in improving compositionality in the case of MosaiCLIP . These two reasons motivate the use of tree-score to compare the language encoder's hierarchical understanding capability. Fig. 4 a) shows that MosaiCLIP's language encoder has higher tree-scores than NegCLIP's language encoder, suggesting that MosaiCLIP performs more tree-like computations. This explains the improved language compositionality of MosaiCLIP since a hierarchical tree-structured computation allows the language encoder to better understand input text compositionally, thereby improving vision-language compositionality. This is in line with the hypothesis that human's semantic understanding of sentences involves a hierarchical (tree-structured) computation which has significant evidence (Crain and Nakayama, 1987; Hale et al.,

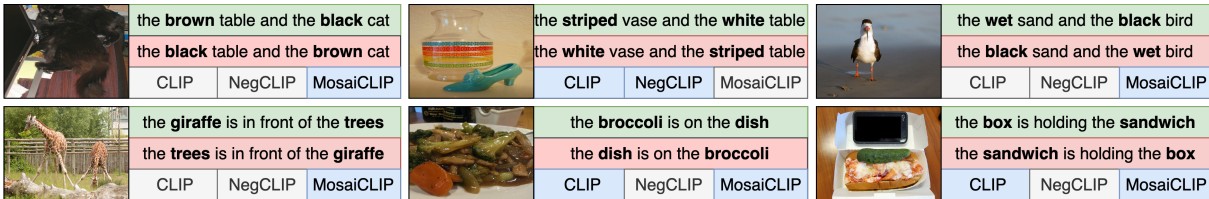

Figure 5: Qualitative analysis on ARO dataset (Top:ARO-Attribution, Bottom: ARO-Relation). Models highlighted in blue match the image to the correct sentence (in green) while the models in white match the image to the incorrect sentence (in red). Here, models are taken from our fine-tuning experiments on COCO from Table 1.

2018; Pallier et al., 2011) and this leads to their compositional generalization capability.

**MosaiCLIP is Robust:** Noisy texts often have meaningful sub-texts which can be exploted by MosaiCLIP, hence MosaiCLIP often achieves consistent performance increase regardless of noise in the pre-training or fine-tuning dataset. For example, NegCLIP achieves significantly low performance on ARO when fine-tuned with YFCC-FT (having more noise in text) as compared CC-FT or COCO as shown in Table 1. NegCLIP takes a $> 10\%$ hit in performance across various ARO datasets when the fine-tuning dataset is changed from COCO to YFCC, whereas, MosaiCLIP achieves similar performance using both datasets. Appendix Sec. D.3 shows that pre-trained MosaiCLIP is robust to natural distributon shifts.

**Qualitative Analysis:** We take MosaiCLIP, NegCLIP and CLIP fine-tuned on COCO and filter out examples from the ARO dataset where MosaiCLIP and NegCLIP's disagree. Some notable examples in Fig. 5 include cases where NegCLIP and CLIP often struggle to understand simple concepts like understanding the color of the cat and table (top-left Fig. 5 or understanding the "is holding" relation b/w sandwich and the box in bottom-right Fig. 5.

### 4.3 Ablations

Table 6 and Appendix Tables 8,9 show the effect of curriculum learning and robust fine-tunining where we find that curriculum learning can bring consistent improvements of up to $1.2\%$ on average and robust-finetuning (WiSE-FT) technique performs the best on zero-shot tasks (i.e. minimal forgetting while fine-tuning), while still improving over NegCLIP by about $5\%$ on compositional reasoning tasks. Table 7 shows the effects of different kinds of sub-graphs sampled during training. More details including the effect of sampling larger number of sub-graphs are presented in Appendix Sec. C.

| Benchmark → | **ARO** | | | **CREPE** | | VLC | SVO | Meta |
|---|---|---|---|---|---|---|---|---|
| Method ↓ | Rel. | Attr. | Ord. | CU | AU | Avg. | Avg. | Avg. |
| **MosaiCLIP** | **80.4** | **69.8** | **85.5** | **72.4** | 40.9 | 77.6 | 88.73 | **73.6** |
| **MosaiCLIP**$_{NoCurric}$ | 79.0 | 69.6 | 80.6 | 71.1 | 40.2 | **77.7** | **88.91** | 72.4 |
| **MosaiCLIP**$_{WiSE-FT}$ | 78.8 | 69.4 | 82.6 | 67.5 | **41.2** | 76.4 | 88.08 | 72.0 |

Table 6: Effect of Curriculum learning and Robust Fine-tuning (**MosaiCLIP**$_{WiSE-FT}$) using CC-FT data.

| Fine-tuning data → | COCO | | CC-FT | | YFCC-FT | |
|---|---|---|---|---|---|---|
| Method ↓ | Rel. | Attr. | Rel. | Attr. | Rel. | Attr. |
| MosaiCLIP | **82.6** | **78.0** | **80.4** | **69.8** | **74.3** | **66.9** |
| without $f_{rel}$ | 81.7 | 76.6 | 78.8 | 68.7 | 73.5 | 66.2 |
| without $f_{attr}$ | 77.7 | 73.2 | 70.5 | 68.2 | 69.0 | 65.9 |
| without $f_{rel}, f_{attr}$ | 79.0 | 70.4 | 68.8 | 64.9 | 57.4 | 63.6 |

Table 7: Effect of different positive-negative sub-graph types sampled while training. Results are presented on the ARO benchmark.

### 5 Conclusion

We present a method to improve the compositional reasoning capabilities of contrastively trained large vision-language models. In particular, we provide a coarse-to-fine contrastive learning framework and a scene graph-based text decomposition strategy for matching subgraphs of the text scene graph having varying complexity to an image during contrastive learning. We also develop hard negative graph creation strategies focused on improving attribute binding and relation understanding capabilities. Our techniques leads to significant improvements in compositional reasoning capabilities. We investigate the reasons for improved compositionality and present a novel finding based on language encoder tree-scores, suggesting that our models learn improved fine-grained and hierarchical text understanding, which is likely the key reason for improved vision and language compositionality of MosaiCLIP as compared to baselines.

## 6 Limitations

Computational Cost: Although MosaiCLIP leads to significant performance increase on several benchmarks that test compositional reasoining, it requires a higher per-batch computational cost while training. For this we give a detailed analysis on the computational cost in Appendix C.6 and show that simply providing more compute to prior methods in the form of larger batch sizes does not improve compositional reasoning. We also show ways to tackle this computational cost, by using less data in Appendix C.5, since MosaiCLIP is data efficient and can provide improvements over baselines with as low as 0.3x of the total data. This along with our ablations in Appendix C.1 gives some control to any practitioner to vary either the training dataset size or the number of sub-graphs in our method, and obtain a clean tradeoff between accuracy and compute. As future work we would like to develop a coarse-to-fine grained objective requiring minimal extra computation cost per batch. Future work should also look at decreasing the extra computational cost incurred by contemporary methods like Syn-CLIP (Cascante-Bonilla et al., 2023) and Teaching SVLC (Doveh et al., 2023).

Other Vision Language Models: In our current work we primarily aim to improve the compositionality of CLIP-Style, dual-tower models trained using large scale *contrastive learning*, since they severely lacked compostional reasoning capabilities as shown by (Yuksekgonul et al., 2022). Many other VLMs exist such as those that undergo cross modal interactions between vision and language such as BLIP (Li et al., 2022b), X-VLM (Zeng et al., 2021), LXMERT (Tan and Bansal, 2019). Although our methods show promise in improving more advanced VLMs like BLIP as shown in Section 4 and Table 5, a more thorough analysis will be beneficial to study the extent to which our methods can improve vision-language contrastive learning for these models.

Sentence Templates: For simplicity, we currently use manually curated templates to convert sub-graphs to sentences, however, this can lead to similar looking and synthetic sentences. Large language models like GPT-4 (OpenAI, 2023), BLOOM (Mitchell et al., May 2021-May 2022) should be looked into for developing sentences from scene-graphs, by directly giving the LLM a scene-graph as input and requiring it to generate a sentence. This approach might be effective but may also lead to higher computational cost while training.

## Acknowledgements

We thank anonymous reviewers for their insightful suggestions that helped in greatly improving our paper. We also thank Aditi Khandelwal, Animesh Sinha, Abhishek Kadian for their helpful comments and suggestions on this work.

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

# Appendix

## A  Background

**Contrastive Language-Image pre-training (Radford et al., 2021)** (CLIP) aims to learn general-purpose representations of vision and language using paired image-text data. This is achieved using contrastive learning in the image-text space. In particular consider a pre-training dataset of size $n$, $\mathcal{D} \subset \mathcal{X} \times \mathcal{T}$, $\mathcal{D} = \{\boldsymbol{x}_i, \boldsymbol{t}_i\}_{i=1}^n$. Here $\mathcal{X}$ and $\mathcal{T}$ are the space of images and text, respectively, and $\boldsymbol{x}_i, \boldsymbol{t}_i$ are images and text in the dataset. Also, consider access to image and text encoders, that we represent by $f_\theta : \mathcal{X} \to \mathbb{R}^d$ and $f_\phi : \mathcal{T} \to \mathbb{R}^d$ respectively. To learn distributed representations for images and text, the following contrastive losses are used:

$$\mathcal{L}_{t2i} = -\frac{1}{|\mathcal{B}|} \sum_{j=1}^{|\mathcal{B}|} \log \frac{\exp(\tau \boldsymbol{u}_i^T \boldsymbol{v}_j)}{\sum_{i=1}^{|\mathcal{B}|} \exp(\tau \boldsymbol{u}_i^T \boldsymbol{v}_j)} \quad (1)$$

$$\mathcal{L}_{i2t} = -\frac{1}{|\mathcal{B}|} \sum_{i=1}^{|\mathcal{B}|} \log \frac{\exp(\tau \boldsymbol{u}_j^T \boldsymbol{v}_j)}{\sum_{j=1}^{|\mathcal{B}|} \exp(\tau \boldsymbol{u}_i^T \boldsymbol{v}_j)} \quad (2)$$

Where $\mathcal{B}$ represents the batch during one iteration of training. $\boldsymbol{u}_i, \boldsymbol{v}_i$ are the $\ell_2$ normalized embeddings of $\tilde{\boldsymbol{u}}_i, \tilde{\boldsymbol{v}}_i$, where $\tilde{\boldsymbol{u}}_i = f_\theta(\boldsymbol{x}_i)$, $\tilde{\boldsymbol{v}}_i = f_\phi(\boldsymbol{t}_i)$. $\tau$ is the temperature parameter and is trainable. The overall loss is $\mathcal{L}_{clip} = (\mathcal{L}_{t2i} + \mathcal{L}_{i2t})/2$.

## B  Scene Graph Decomposition

Here we provide additional details for text scene graph decomposition. Denote the text scene graph obtained from the scene graph parser by $G_T = (V_T, E_T)$, where $V_T$ represent the nodes of the graph, which are either objects or their attributes. $E_T$ are the edges of the graph that represent relations between objects. Let $\mathbb{G}$ denote the set of all possible scene graphs. We first consider an external set of objects ($\mathcal{N}$), attributes ($\mathcal{A}$), and relations ($\mathcal{R}$) that we use for creating negative sub-graphs. In practice, we create this set from Visual Genome (VG) dataset (Krishna et al., 2016). Following Zhang et al. (2021), we sample a total of 1594 entities that have 30 instances of them in the VG dataset. The attribute and Relation list contains 524, and 50 unique instances, respectively. Hence $|\mathcal{N}| = 1594$, $|\mathcal{A}| = 524$, $|\mathcal{R}| = 50$. We first sample all possible sub-graphs having *one* or *two*

objects in them, and these can have multiple attributes for the objects. We develop and use scene graph transformations that take a sub-graph as input and return a (set of) modified versions of the graph (minimally-perturbed negative sub-graphs for the image). For this, we define three graph transformations as follows:

- $f_{obj} : \mathbb{G} \longrightarrow P(\mathbb{G})$ takes input a single object scene graph, where the object has attributes $A_o$. For each attribute, $a \in A_o$, a random attribute $a'$ is sampled uniformly at random from $\mathcal{A}$. We finally obtain a set of sub-graphs $G_{obj} \in P(\mathbb{G})$ where $P(.)$ denotes the power set. Each $g \in G_{obj}$ contains one object node connected with an attribute node which is sampled from $\mathcal{A}$.

- $f_{rel} : \mathbb{G} \longrightarrow P(\mathbb{G})$ takes input sub-graphs having one relation edge and gives output a set of sub-graphs $G_{rel} \in P(\mathbb{G})$ where each $g \in G_{rel}$ has either object nodes shuffled, replaced by an external object node $n'$ sampled uniformly at random from $\mathcal{N}$, and/or relation replaced by external relation $r'$ sampled uniformly at random from $\mathcal{R}$. Along with this, we also *join* the input positive sub-graph with a random sub-graph created by sampling random nodes and edges from $\mathcal{N}, \mathcal{A}, \mathcal{R}$.

- $f_{attr} : \mathbb{G} \longrightarrow P(\mathbb{G})$ takes input sub-graphs having one relation edge and gives output a set of sub-graphs $G_{attr} \in P(\mathbb{G})$ where each $g \in G_{attr}$ has attribute nodes shuffled, and/or replaced by an external attribute node $a'$ sampled uniformly at random from $\mathcal{A}$.

$f_{obj}, f_{attr}$ broadly aims at improving the model's attribute understanding, while $f_{rel}$ broadly targets improved relation understanding. For each positive sub-graph, we sample all possible negative subgraphs using $f_{obj}, f_{rel}, f_{attr}$ and make positive-negative sub-graph pairs $(g_{pos_i}, \{g_{neg_i}\})$. These pairs can be classified into three categories $C = \{c_{obj}, c_{rel}, c_{attr}\}$ according to the transformation that created the negative sub-graphs. We sample sub-graph pairs from these categories according to probabilities $p_i$, $i \in \{1, 2, 3\}$ corresponding to the three categories respectively, and $\sum p_i = 1$. These probabilities are hyperparameters; see Appendix Section H.1 for more details. Multiple sub-graph pairs can have common positive or negative sub-graphs, and sampling these pairs would result

in duplication, hence for each image, we make sure to deduplicate sub-graphs so that all sub-graphs, and therefore the text made from them are unique for a given image in a batch. After sampling, all sub-graphs are transformed to text using simple templates, as explained in Section 3.3.

## C   Ablations and Model Analysis

### C.1   Sampling more subgraphs

We analyze the effect of increasing the maximum number of sub-graphs sampled for any given image in a batch of data during training. See Figures 6 and 7, in which we test the performance on ARO and CREPE benchmarks (averaged over three fine-tuning datasets considered in this work), as we increase the max positive and negative sub-graphs per image. We find that as we increase both positive and negative sub-graphs for an image, the performance steadily increases up to a point for *all* datasets, after which the performance can either flatten out, increase, or even decrease in some of the datasets. This is intuitive since a larger number of positive and negative sub-graphs per image leads to a gap w.r.t the pre-training stage as described in Sec. 3.6. Also, different compositional splits require different reasoning skills, and as we keep sampling positive and negative sub-graphs for an image, it is natural for certain types of positive and negative sub-graphs to be more pronounced, depending on the dataset statistics, and this can have varied effects on different datasets.

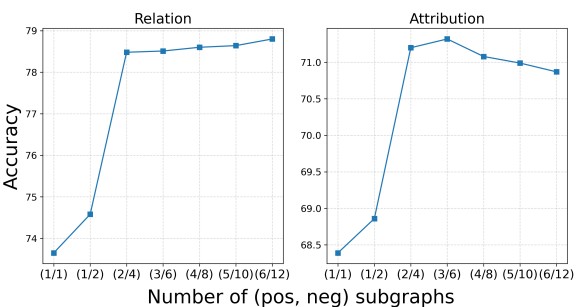

Figure 6: Effect of increasing the number of positive and negative subgraphs on ARO benchmark when fine-tuning MosaiCLIP. Results are averaged over 3 fine-tuning datasets considered in this work

### C.2   Effect of different sub-graph types

Here we analyze the effect of sampling different kinds of sub-graphs from the original scene graph of the text. In particular, we measure the effect of graph transformations that we define in Appendix

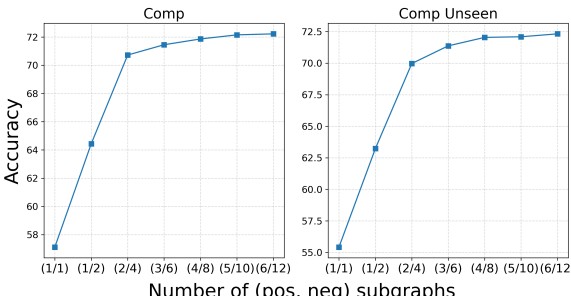

Figure 7: Effect of increasing the number of positive and negative subgraphs on CREPE - Systematicity benchmark, when fine-tuning MosaiCLIP (Here we use Open-CLIP RN-50 model pre-trained on CC-12M and fine-tune it on CC-FT).

Sec. B. Results are presented in Table 7. We observe that both $f_{rel}$ and $f_{attr}$ as described in Appendix Sec. B, are useful for improving relation and attribute understanding (as measured on the ARO benchmark), across fine-tuning datasets.

### C.3   Effect of curriculum training

As shown in Table 8, in all fine-tuning results, we can see consistent improvements when using our curriculum learning strategy, such as upto $2\%$ on systematic generalization, and sometimes more than $6\%$ as seen for ARO-Order results when the fine-tuning dataset is YFCC-FT.

### C.4   Effect of robust fine-tuning

Among many other techniques developed for mitigating forgetting in large models when they are fine-tuned, one prominent one is robust fin-tuning-WiSE-FT, (Wortsman et al., 2022). Following Wortsman et al. (2022) we perform weight-space ensembling on the image encoder before and after fine-tuning using our method and call this model MosaiCLIP$_{\text{WiSE-FT}}$. The results on compositionality benchmarks can be seen in Table 8 while results on 21 multimodal tasks from ELEVATER and ImageNet can be seen in Table 9. We find that MosaiCLIP$_{\text{WiSE-FT}}$ has a slight performance decrease on some compositonal benchmarks as compared to MosaiCLIP, however, it is significantly better than NegCLIP on most benchmarks. The real benefit of using MosaiCLIP$_{\text{WiSE-FT}}$ is that it leads to least forgetting, and there is little to no performance degradation on 21 tasks as showin in Table 9.

| FineTun. data → | COCO | | | | | CC-FT | | | | | | | | YFCC-FT | | | | | | | |
|---|---|---|---|---|---|---|---|---|---|---|---|---|---|---|---|---|---|---|---|---|---|---|
| Benchmark → | ARO | | | VLC | SVO | ARO | | | CREPE | | VLC | SVO | | ARO | | | CREPE | | VLC | SVO | Meta |
| Method ↓ | Rel. | Attr. | Ord. | Avg. | Avg. | Rel. | Attr. | Ord. | CU | AU | Avg. | Avg. | | Rel. | Attr. | Ord. | CU | AU | Avg. | Avg. | Avg. |
| CLIP | 59.8 | 63.2 | 53.3 | 70.8 | 83.58 | 59.8 | 63.2 | 53.3 | 45.1 | 35.0 | 70.8 | 83.58 | | 59.8 | 63.2 | 53.3 | 39.8 | 39.5 | 70.8 | 83.58 | 60.60 |
| NegCLIP | 81.7 | 72.7 | 85.7 | 75.6 | 90.20 | 71.5 | 65.4 | 84.5 | 53.1 | 37.5 | 72.4 | 88.36 | | 57.8 | 63.1 | 52.1 | 38.8 | 39.0 | 70.4 | 83.90 | 67.57 |
| **MosaiCLIP** | **82.6** | **78.0** | 87.1 | **81.4** | **90.67** | **80.4** | **69.8** | **85.5** | **72.4** | **40.9** | 77.6 | 88.73 | | **74.3** | 66.9 | **84.4** | **48.8** | 41.5 | 75.1 | **85.36** | **74.29** |
| **MosaiCLIPNoCurric** | 81.6 | 76.8 | **87.4** | **81.4** | 90.20 | 79.0 | 69.6 | 80.6 | 71.1 | 40.2 | **77.7** | **88.91** | | 74.1 | **67.2** | 77.8 | 46.6 | 40.5 | 75.7 | 84.97 | 73.23 |
| **MosaiCLIPWiSE-FT** | 82.5 | 76.2 | 86.6 | 80.3 | 89.65 | 78.8 | 69.4 | 82.6 | 67.5 | 41.2 | 76.4 | 88.08 | | 69.4 | 67.0 | 79.4 | 48.1 | 43.6 | 74.2 | 83.71 | 72.88 |

Table 8: Ablating the effect of Curriculum learning and Robust fine-tuning. **MosaiCLIPNoCurric** refers to the version of our model without any curriculum learning. **MosaiCLIPWiSE-FT** refers to the version where the image encoder of the final model (after fine-tuning) and before fine-tuning are weight-space ensembled. CLIP and NegCLIP scores are also shown for reference. See Appendix Sec. C.3.

| Method | ZS(21) | Compositional Score (Meta Avg.) |
|---|---|---|
| CLIP | **56.4** | 60.60 |
| NegCLIP | 56.8 | 67.57 |
| MosaiCLIPNoCurric | 55.8 | 73.23 |
| MosaiCLIPWiSE-FT | 56.8 | 72.88 |
| MosaiCLIP | 55.7 | **74.29** |

Table 9: Zero Shot accuracy on 21 multimodal datasets from ELEVATER and ImageNet. Results are average of the three fine-tuning datasets. MosaiCLIP has negligible drop in performance in general (compared to the gains on compositionality benchmarks), and one can boost performance by using MosaiCLIPWiSE-FT which has equal performance as compared to NegCLIP on 21 muldimodal datasets. Meta Avg. Compositional Score is taken from Table 8. Second best results are underlined. **Conclusion:** One can use MosaiCLIP for getting the best compositional reasoning capabilities with minimal performance degradation on multimodal tasks, and use MosaiCLIPWiSE-FT for no degradation in performance on multimodal tasks, while still performing well on compositional reasoning.

| Method | Fraction of data | ARO | | SVO |
|---|---|---|---|---|
| | | Rel. | Attr. | Avg. |
| NegCLIP | Full | 73.6 | 58.9 | 76.10 |
| MosaiCLIP | 0.3x | 71.6 | 60.6 | 70.82 |
| | 0.5x | 74.3 | 60.8 | 74.04 |
| | 0.6x | 74.7 | 63.8 | 75.76 |
| | 0.8x | 77.0 | 66.3 | 77.22 |
| | Full | 74.7 | 66.1 | 77.87 |

Table 10: Data efficiency of MosaiCLIP during pre-training. Numbers in blue are lowest numbers that are within 1% or greater than NegCLIP performance. Pre-Training dataset: YFCC-15M.

| Method | Fraction of data | ARO | | SVO |
|---|---|---|---|---|
| | | Rel. | Attr. | Avg. |
| NegCLIP | Full | 71.5 | 65.4 | 88.36 |
| MosaiCLIP | 0.3x | 70.8 | 67.7 | 88.70 |
| | 0.5x | 74.5 | 68.6 | 88.80 |
| | 0.6x | 75.3 | 69.3 | 88.76 |
| | 0.8x | 78.2 | 69.8 | 88.98 |
| | Full | 79.0 | 69.6 | 88.91 |

Table 11: Data efficiency of MosaiCLIP during fine-tuning. Numbers in blue are lowest numbers that are within 1% or greater than NegCLIP performance. Fine-tuning dataset: CC-FT. Curriculum learning has not been used for these experiments.

## C.5 Data efficiency

We find that our technique leads to significant data efficiency requiring about 0.3x-0.6x fo the total fine-tuning or pre-training data to match or exceed NegCLIP performance. Results are shown in Tables 10 and 11.

## C.6 Computational cost

Even though MosaiCLIP uses the same global batch size of image-text pairs, it requires more compute as compared to NegCLIP or CLIP owing to the fact that decomposing sub-graph leads to a larger effective text-batch size and hence a larger contrastive learning matrix. It is a common practice in literature to trade-off larger compute for improving

CLIP's compositionality, as also done by previous methods Syn-CLIP (Cascante-Bonilla et al., 2023) that generate data using external graphics engines, and Teaching-SVLC (Doveh et al., 2023) which use LLMs requiring massive compute even during inference.

**Providing NegCLIP with more compute:** One can argue that providing more compute to NegCLIP can lead to better performance, however, on the contrary we found that NegCLIP's performance

decreases as batch size is scaled (from 256 to 4096, much beyond MosaiCLIP's text or image batch size), as shown in Table 12.

**Performance-Compute Tradeoff:** It is to be noted that MosaiCLIP performance continues to increase up to a threshold, as sub-graphs are increased as shown in Table 7 and 6 hence this provides a clean tradeoff between number of sub-graphs and compute, and a practitioner can choose the number of sub-graphs their compute availablility. Along with this, in Appendix Sec. C.5 we showed that we can achieve improved performance compared to NegCLIP with as low as 0.3x data closing the gap between NegCLIP and MosaiCLIP compute even more. *It is to be noted that MosaiCLIP is a drop in replacement for CLIP after training and requires the same inference cost as CLIP.*

| Batch Size (B) | ARO | | SVO |
|---|---|---|---|
| | Rel. | Attr. | Avg. |
| 512 | 68.9 | 65.6 | 88.68 |
| 1024 | 67.6 | 65.1 | 88.93 |
| 2048 | 65.7 | 64.2 | 88.72 |
| 4096 | 62.5 | 63.7 | 88.11 |

Table 12: Performance of NegCLIP with increasing batch size. A batch size of B corresponds to an effective batch size of 8*B in NegCLIP after image and text negative mining. Fine-tuning dataset: CC-FT.

# D   Additional Results and Experiments

## D.1   Comparison with recent baselines

We compare with recently published and contemporary works (Cascante-Bonilla et al., 2023; Doveh et al., 2023). Doveh et al. (2023) show that one can create rule-based hard negative sentences and Large Language Models (LLMs) based hard negative sentences and use them when training CLIP style models to obtain an improved model that is better at handling tasks that require compositional reasoning. We fine-tune on CC3M (Sharma et al., 2018) for a fair comparison with Doveh et al. (2023). Results are reported in Table 13. A fair comparison with Syn-CLIP Cascante-Bonilla et al. (2023) is not possible since their synthetic dataset is not released. However in Table 13 we find that performance difference is large between MosaiCLIP and Syn-CLIP showing that our general coarse-to-fine grained approach is better than using targeted synthetic datasets for inducing compositional understanding in VLMs. Comparisons with Doveh et al. (2023) in

Table show that our approach is competitve or better at attribute, relation and object understanding as measured by the VL-Checklist benchmark (Zhao et al., 2022). Zero Shot performance on 21 datasets suffers minimally using our approach, and is even better than (Zhao et al., 2022). It is to be noted that both approaches Syn-CLIP (Cascante-Bonilla et al., 2023) and Doveh et al. (2023) are orthogonal to our approach and combining them with our coarse-to-fine understanding approach will likely result in much better performance overall, as compared to individual techniques. In particular, Syn-CLIP (Cascante-Bonilla et al., 2023) faces the issue of having long captions for images, and they average out embeddings of parts of the caption before matching it to the image. This issue can be eaily resolved using our framework which can easily handle multiple positive captions for an image. Performing this ablation would be future work for us, once synthetic datasets like that used by Cascante-Bonilla et al. (2023) are open-sourced and gain more popularity. Our approach can similarly also include captions generated from LLMs, as explored by Doveh et al. (2023).

## D.2   Standard deviations for fine-tuning results

Here we provide fine-tuning results on the CC-FT dataset *with standard deviations* over 3 random seeds where OpenAI CLIP-ViT-B-32 is fine-tuned on CC-FT using MosaiCLIP and baseline techniques. See Table 14 for the results. The main paper Table 1 have average results for CC-FT while for COCO and YFCC-FT fine-tuning datasets, the results are for one seed. We do-not run multiple pre-training experiments since they significantly more costly.

## D.3   Robustness to natural distribution shifts

We find that pre-trained MosaiCLIP shows robustness to natural distribution shifts as measured by ImageNet natural distribution shifts benchmark. Results are presented in Table 15. We believe that MosaiCLIP sees a larger variety of texts in the form of sub-graphs which can provide it with extra supervision for tackling natural distribution shifts. Intutively, sub-graphs can lead to diversity of texts being seen by the model during training and this might lead to broader coverage of concepts and concept combinations, resulting in improved robustness. Along with this a coarse to fine hierarchical understanding of texts and thereby, of

| Benchmark → | VL-Checklist | | | ARO | | | ZS(21) |
|---|---|---|---|---|---|---|---|
| Method | Obj. | Attr. | Rel. | Rel. | Attr. | Ord. | Avg. |
| CLIP | 81.6 | 67.6 | 63.1 | 59.9 | 63.6 | 53.3 | **56.4** |
| CLIP-FT | 79.0 | 64.7 | 54.3 | 41.7 | 59.3 | 25.2 | 56.9 |
| Syn-CLIP† (Cascante-Bonilla et al., 2023) | - - | 70.4 | 69.4 | 71.4 | 66.9 | 65.1 | 55.3 |
| Teaching SVLC‡ (Doveh et al., 2023) | 85.0 | 72.0 | 69.0 | - - | - - | - - | 54.8 |
| **MosaiCLIP$_{NoCurric}$** | 86.4 | **75.0** | 69.6 | 83.2 | **78.6** | 77.3 | 54.9 |
| **MosaiCLIP$_{WiSE-FT}$** | **86.5** | 73.6 | **72.2** | 82.6 | 77.0 | **79.9** | 55.9 |
| **MosaiCLIP** | 86.4 | 73.7 | 71.9 | **83.7** | 78.0 | 79.4 | 53.5 |

Table 13: Comparison of MosaiCLIP with recently published and contemporary works Syn-CLIP (Cascante-Bonilla et al., 2023) and Teaching SVLC Doveh et al. (2023). Results are reported on VL-Checklist, ARO and Average Zero Shot results on 21 datasets from ELEVATER and Imagenet. Performance numbers of these models are reported from their respective papers (blank fields (—) are not reported in respective papers). †Uses million-scale synthetic data for fine-tuning. ‡Uses external Large Language Models (LLMs) like BLOOM (Mitchell et al., May 2021-May 2022) for text augmentation and hard negative text creation. See Sec. D.1 for more details.

| Benchmark → | ARO | | | SVO-Probes | | |
|---|---|---|---|---|---|---|
| Method ↓ | Rel. | Attr. | Ord. | Obj. | Subj. | Verb. |
| CLIP-FT | 58.1±0.63 | 63.3±0.28 | 42.7±0.18 | 93.17±0.11 | 88.64±0.17 | 83.87±0.03 |
| NegCLIP | 71.5±0.40 | 65.4±0.58 | 84.5±0.11 | 92.90±0.09 | 88.16±0.11 | 84.02±0.02 |
| **MosaiCLIP$_{NoCurric}$** | 79.0±0.66 | 69.6±0.19 | 80.6±0.17 | 93.37±0.04 | 89.74±0.13 | 83.62±0.04 |
| **MosaiCLIP** | 80.4±0.63 | 69.8±0.21 | 85.5±0.16 | 93.45±0.04 | 89.39±0.07 | 83.35±0.05 |

Table 14: Fine-Tuning Results on CC-FT dataset *with standard deviations* across 3 random seeds. These results correspond to the CC-FT fine-tuning results in main paper Table 1. Here the base model which is fine-tuned using different techniques is OpenAI-CLIP-ViT-B-32.

images should intuitively help in improving performance on robustness benchmarks given that the model will now be able to recognise details in images and texts more accurately.

# E  Dataset Details

Here we provide deteiles about datasets used for fine-tuning, pre-training and evaluating models in this study. A summary is shown in Table 16

## E.1  Fine-tuning datasets

Following NegCLIP (Yuksekgonul et al., 2022) we use the COCO dataset released by (Yuksekgonul et al., 2022) having 109k samples that had hard negative sentences that (Yuksekgonul et al., 2022) create for training NegCLIP. As mentioned in the main paper, COCO dataset images are used for creating Visual Genome (Krishna et al., 2016), and this is further used to create datasets such as CREPE (Ma et al., 2022), ARO (Yuksekgonul et al., 2022) and a part of VL-Checklist (Zhao et al., 2022). This can lead to confounding and potentially mislead-

ing results, since it is unclear if the performance increase using any method comes from the fine-tuning dataset (COCO) being close to the domain of test datasets, or if it's the fine-tuning methodology that leads to an increase in performance. Hence, for rigourous experimentation of the developed methods, one must use other datasets to fine-tune contrastively trained VLMs. We randomly sample similar sized (100k datapoints) from popular pre-training datasets CC-12M and YFCC-15M, and call these smaller datasets CC-FT and YFCC-FT. To train NegCLIP, hard negative sentences and images are required, for which we first use the code released by (Yuksekgonul et al., 2022)[2] to create hard negatives sentences as well as sample three hard negative images for each image based on OpenAI CLIP ViT-B/32 features, strictly following (Yuksekgonul et al., 2022). For comparing with contemporary works (Doveh et al., 2023), (Cascante-Bonilla et al., 2023) (as shown in Table

[2] https://github.com/mertyg/vision-language-models-are-bows

| Arch. | Data | Method | ImageNet-A | | ImageNet-R | | ImageNet-S | | ImageNet-V2 | |
|---|---|---|---|---|---|---|---|---|---|---|
| | | | Top1 | Top5 | Top1 | Top5 | Top1 | Top5 | Top1 | Top5 |
| Swin-T | CC-12M | CLIP | 6.4 | 24.5 | 42.6 | 68.8 | 22.2 | 45.5 | 28.2 | 54.1 |
| | | NegCLIP | 6.6 | 25.0 | 43.1 | 68.7 | 22.2 | 45.4 | 29.4 | 55.2 |
| | | **MosaiCLIP** | **9.1** | **29.4** | **48.6** | **74.3** | **27.2** | **52.6** | **33.6** | **61.6** |
| | YFCC-15M | CLIP | 10.9 | 34.2 | 20.6 | 42.0 | 6.4 | 16.7 | 26.1 | 49.9 |
| | | NegCLIP | 11.4 | 35.6 | 20.0 | 41.7 | 6.0 | 16.0 | 27.2 | 50.7 |
| | | **MosaiCLIP** | **14.6** | **40.2** | **22.3** | **44.9** | **6.8** | **17.7** | **32.0** | **57.2** |
| RN-50 | CC-12M | CLIP | 7.3 | 27.4 | 41.4 | 67.8 | 21.7 | 44.3 | 29.8 | 56.4 |
| | | NegCLIP | 7.7 | 27.7 | 41.0 | 66.9 | 21.7 | 43.9 | 30.2 | 56.0 |
| | | **MosaiCLIP** | **11.1** | **35.6** | **52.1** | **76.9** | **29.5** | **55.4** | **37.0** | **66.5** |
| | YFCC-15M | CLIP | 13.4 | 37.3 | 17.2 | 37.2 | 4.9 | 13.6 | 25.8 | 49.4 |
| | | NegCLIP | 12.9 | 38.0 | 18.0 | 37.3 | 5.1 | 14.7 | 26.0 | 49.0 |
| | | **MosaiCLIP** | **17.4** | **46.6** | **21.0** | **42.7** | **6.5** | **16.9** | **32.2** | **57.9** |

Table 15: Results on ImageNet - Natural Distribution Shifts datasets. **MosaiCLIP** leads to improved robustness to natural distribution shifts. NegCLIP performs similarly as CLIP. Models are zero-shot tested on ImageNet-A (Hendrycks et al., 2021b), ImageNet-R (Hendrycks et al., 2021a), ImageNet-S(ketch) (Wang et al., 2019) and ImageNet-V2 (Recht et al., 2019).

| Benchmark/Dataset | #Examples | #Subtasks | Subtask Examples | Datasets Used for Creation |
|---|---|---|---|---|
| Compositional Reasoning (Evaluation) | | | | |
| ARO | 77K | 3 | Attribute, Relation, Order understanding | Visual Genome, COCO, Flickr |
| CREPE-Systematicity | 642K | 2 | Systematic generalization generalization | Visual Genome |
| VL-Checklist | 410K | 3 | Attribute, Relation Object understanding | Visual Genome HAKE, VAW, SWiG |
| SVO-Probes | 48K | 3 | Verbs (Relations) understanding | – |
| CREPE-Productivity | 183K | 9 | Productivity | Visual Genome |
| Fine-Tuning datasets | | | | |
| COCO | 109K | – | – | – |
| CC-FT | 100K | – | – | – |
| YFCC-FT | 100K | – | – | – |
| CC-3M | 3.11M | – | – | – |
| Pre-Training datasets | | | | |
| CC-12M | 11.26M | – | – | – |
| YFCC-15M | 14.20M | – | – | – |

Citations: ARO(Yuksekgonul et al., 2022), CREPE(Ma et al., 2022), VL-Checklist(Zhao et al., 2022), SVO(Hendricks and Nematzadeh, 2021), Visual Genome(Krishna et al., 2016), COCO(Lin et al., 2014), Flickr(Young et al., 2014), HAKE(Li et al., 2019), VAW(Pham et al., 2021), SWiG(Pratt et al., 2020)

Table 16: Details of datasets used in this study for testing compositional reasoning, for fine-tuning and pre-training models. See Appendix Sec. E for more details.

2), we use CC3M (Sharma et al., 2018) since it's used by these baselines, and makes a direct comparison possible with them.

## E.2 Pre-training datasets

We use popular and standard large scale pre-training datasets CC-12M (Changpinyo et al., 2021)

and YFCC-15M (Thomee et al., 2016) for pre-training all models in this study, including CLIP, NegCLIP and MosaiCLIP.

### E.3 Evaluation datasets

Here we list the evaluation detailes used in this study and also provide a short description for each **CREPE-Systematicity** (Ma et al., 2022): CREPE provides systematic generalization datasets to test models trained on popular pre-training datasets including CC-12M and YFCC-15M. While creating CREPE, Ma et al. (2022) make sure to split the dataset into seen and unseen parts, which correspond to weather the model has seen or not seen the combination of concepts, when pre-trained with popular pre-training datasets. We measure and report performance on both seen and unseen splits in our work.

**ARO** (Yuksekgonul et al., 2022): This benchmark consists of four datasets, including VG-Relation, VG-Attribution, COCO-Order, and Flickr-Order. The first two measure attribute and relation understanding of VL models, respectively, and the last two measure the word order understanding of VL models. VG-Relation and VG-Attribution consist of tuples having an image and two texts (one positive and one negative), and the model's task is to match the image with the correct text. order datasets have four negative texts and one positive text for each image, and the task is again to match the image with the correct text.

**SVO-Probes** (Hendricks and Nematzadeh, 2021): This dataset consists of tuples having two images and one text. All texts and images have a subject, verb, and object, and the images differ in only one of subject, verb, or object. This dataset helps in understanding if VL models can compositionally understand combinations of objects having a relation between them. The original dataset contains 48K examples.[3]

**CREPE-Productivity** (Ma et al., 2022): Productivity dataset tests the model's ability to generalize to longer and more complex sentences, with complexity ranging from 4 atoms to 12 atoms, where an atom can be an attribute, relation, or object. The CREPE-Productivity dataset has a number of test sets for each sentence complexity ranging from 4 atoms to 12 atoms.

---

[3]Some image links provided by the the original repository(https://github.com/deepmind/svo_probes) were broken. In total, 36k data points were retrievd and used in this study.

**VL-Checklist** (Zhao et al., 2022): This benchmark is created by combining annotations from datasets like Visual Genome (Krishna et al., 2016), SWiG (Pratt et al., 2020), HAKE (Li et al., 2019), VAW (Pham et al., 2021). Each image in the resulting dataset has two captions, a positive and a negative. The positive caption is taken from the source dataset of the image, while the negative caption differs from the positive in only one word which makes it a hard negative and helps in testing compositional and fine-grained understanding of VLMs across various dimensions like attributes, relations, and size and locations of objects.

## F Baselines:

Here we list the baselines used in this study and also provide a short description for each. **CLIP**(Radford et al., 2021): Our first baseline is CLIP model released by OpenAI CLIP(Radford et al., 2021) and OpenCLIP (Ilharco et al., 2021). In particular we use the ViT-B/32 model for fine-tuning results Table 1 of the main paper, except for CREPE dataset, which requires using models pre-traoined on specific datasets, for which we use ResNet-50 (RN-50) models pre-trained on CC-12M and YFCC-15M released by OpenCLIP repository[4] (Ilharco et al., 2021).

**CLIP-FT**: For disentangling the effects of fine-tuning data, and fine-tuning methodology, we create a CLIP-FT baseline where we simply fine-tune the pre-trained CLIP model on the dataset at hand, by using the standard contrastive learning technique used by CLIP.

**NegCLIP**(Yuksekgonul et al., 2022) [ICLR 2023]: NegCLIP is trained using negative mining of texts and images. Yuksekgonul et al. (2022) create sentence level hard negatives by swapping different linguistic elements. They also additionally include hard-negative images and their corresponding texts in the batch by fetching K nearest neighbours (K=3) for each image in the feature space constructed using a pretrained CLIP model.

**Teaching SVLC**(Doveh et al., 2023) [CVPR 2023]: This method uses LLM's like BLOOM (Mitchell et al., May 2021-May 2022) along with rules to create additional positive and negative sentences for each image while fine-tuning CLIP.

**Syn-CLIP**(Cascante-Bonilla et al., 2023) [Arxiv

---

[4]https://github.com/mlfoundations/open_clip

2023]: Syn-CLIP uses a million scale synthetic dataset to fine-tune CLIP and improve it's performance on compositional reasoning tasks. The synthetic data is created using a 3D physics-based simulation platform built on Unity3D, called ThreeDWorld (Gan et al., 2021). This contemporary work is complementary to our data-centric approach and we believe our methods can help fine-tuning with synthetic datasets as well. Cascante-Bonilla et al. (2023) in their paper showed how dense and long captions can be obtained for synthetic images and which require splitting into sub-captons followed by averaging of features from all captions while fine-tuning CLIP. This is one avenue where we believe our method can be useful since our method inherently allows matching of images to multiple texts. This is part of future work, once such synthetic datasets are released and are easily available.

## G   Detailed Experimental Results

In the main paper Table 1 and Table 3 we had provided concise results for some datasets, based on lack of space due to extensive experimental results. Here we provide detailed results on these datasets:

### G.1   VL-Checklist: detailed results

Detailed Fine-tuning results on VL-Checklist dataset are provided in Table 17. These are an extension to the VL-Checklist results provided in the main paper Table 1. Detailed Pre-training results for VL-Checklist dataset are provided in Table 18 which are an extension to the VL-Checklist results provided in the main paper Table 3.

### G.2   SVO-Probes: detailed results

Detailed Fine-tuning results on SVO-Probes dataset are provided in Table 19. These are an extension to the SVO-Probes results provided in the main paper Table 1. Detailed Pre-training results for SVO-Probes dataset are provided in Table 20 which are an extension to the SVO-Probes results provided in the main paper Table 3.

### G.3   CREPE-Systematicity: detailed results

Here we provide detailed results on CREPE-Systematicity dataset used for measuring systematic generalization. In the main paper we had only provided the results related to systematic generalization (i.e., the unseen split), but here we provide results on both the seen and unseen split, for both hard negative retrieval sets (Comp and Atom) that are used when evaluating performance on CREPE by Ma et al. (2022). Detailed Fine-tuning results on CREPE-Systematicity dataset on both the seen and unseen splits are provided in Table 21. These are an extension to the CREPE-Systematicity results provided in the main paper Table 1. Detailed Pre-training results for CREPE-Systematicity dataset are provided in Table 22 which are an extension to the CREPE-Systematicity results provided in the main paper Table 3.

## H   Reproducibility

Here we provide necessary details to reproduce our work, that might not have been included in the main paper.

### H.1   Training and hyperparameter details

Fine-tuning: For all fine-tuning experiments, we follow Yuksekgonul et al. (2022) for hyperparameters. In particular, all models are fine-tuned for 5 epochs, with a batch size of 256, using a cosine learning rate schedule with 50 steps of warmup and random-crop augmentation during training. AdamW is used for optimization. $1e-5$ is used as the initial learning rate. Training is performed using 4 NVIDIA A100 GPUs for all models. From the ARO dataset, $10\%$ examples from attribute and relation splits are used as validation examples, and the rest are used as the test set for all models. On all other datasets, we evaluate zero-shot performance. For MosaiCLIP, we find that sampling a maximum of 3 positive and 6 negative sub-graphs per image during fine-tuning gives the best result on the ARO validation set and hence is used in all our experiments (including pre-training experiments). For MosaiCLIP, we keep sub-graph sampling probabilities as $p_2 = p_3$. We vary $p_1$ in $\{0, 0.08, 0.15\}$ while fine-tuning on the randomly chosen YFCC dataset. We choose the best model according to the ARO val-set and keep the hyperparameters the same for all other fine-tuning datasets.

Pre-training: For pre-training experiments, we follow the training protocol used in Yang et al. (2022); Radford et al. (2021). In particular, all models are trained for 32 epochs, with a batch size of 4096, using a cosine learning rate schedule with 5000 steps of warmup and random-crop augmentation during training. AdamW is used for optimization. The initial learning rate is $1e-3$, and weight decay is

| Benchmark → | VL-Checklist | | | | | | | | |
|---|---|---|---|---|---|---|---|---|---|
| Fine-tuning data → | CC-100K | | | YFCC-100K | | | COCO | | |
| Method | Obj. | Attr. | Rel. | Obj. | Attr. | Rel. | Obj. | Attr. | Rel. |
| CLIP | 81.6 | 67.6 | 63.1 | 81.6 | 67.6 | 63.1 | 81.6 | 67.6 | 63.1 |
| CLIP-FT | 81.9 | 69.3 | 60.9 | 80.7 | 68.1 | 60.2 | 83.7 | 66.7 | 59.3 |
| NegCLIP | 82.1 | 71.4 | 70.3 | 81.0 | 68.1 | 67.1 | 85.2 | 67.2 | 63.0 |
| **MosaiCLIP**$_{\text{NoCurric}}$ | **86.0** | **77.2** | 77.7 | 84.0 | **72.2** | 75.1 | **89.2** | 70.4 | 72.6 |
| **MosaiCLIP**$_{\text{WiSE-FT}}$ | 85.3 | 71.4 | 72.4 | 83.6 | 69.5 | 69.6 | 88.5 | **75.5** | **77.0** |
| **MosaiCLIP** | **86.0** | 76.8 | **78.4** | **84.1** | 72.1 | 74.8 | 89.0 | 70.1 | 71.3 |

Table 17: Fine-tuning results on the VL-Checklist benchmark, for testing compositionality in terms of attribute, relation and object understanding. OpenAI CLIP VIT-B-32 pre-trained model is used as the base model for fine-tuning. See Sec. G.1 for more details.

| Benchmark → | | VL-Checklist | | | | | |
|---|---|---|---|---|---|---|---|
| Pre-training data → | | CC-12M | | | YFCC-15M | | |
| Arch. | Method | Obj. | Attr. | Rel. | Obj. | Attr. | Rel. |
| Swin-T | CLIP | 75.2 | 61.1 | 60.6 | 73.6 | 63.0 | 62.0 |
| | NegCLIP | 75.0 | 67.7 | **67.4** | 71.2 | 66.5 | 60.3 |
| | **MosaiCLIP** | **80.0** | **72.9** | 64.4 | **79.3** | **71.3** | **64.8** |
| RN-50 | CLIP | 75.5 | 62.7 | 60.5 | 73.2 | 62.8 | 58.3 |
| | NegCLIP | 75.4 | 67.6 | **65.5** | 72.9 | 65.8 | 59.7 |
| | **MosaiCLIP** | **79.2** | **73.2** | 65.3 | **80.1** | **71.6** | **65.1** |

Table 18: Pre-training results on VL-Checklist benchmark, for testing compositionality in terms of attribute, relation and object understanding. Results for both backbones Swin-Tiny and RN-50 are shown. See Sec. G.1 for more details.

| Benchmark → | | | SVO-Probes | | | |
|---|---|---|---|---|---|---|
| Arch. | Data | Method | Obj | Subj | Verb | Avg |
| Swin-T | CC-12M | CLIP | 88.43 | 82.58 | 79.33 | 82.21 |
| | | NegCLIP | 88.38 | 81.83 | 79.40 | 82.04 |
| | | **MosaiCLIP** | **91.89** | **87.11** | **82.20** | **85.62** |
| | YFCC-15M | CLIP | 83.38 | 77.09 | 72.80 | 76.27 |
| | | NegCLIP | 84.07 | 76.87 | 72.28 | 76.10 |
| | | **MosaiCLIP** | **86.20** | **79.24** | **73.61** | **77.87** |
| RN-50 | CC-12M | CLIP | 87.86 | 82.54 | 79.45 | 82.13 |
| | | NegCLIP | 87.58 | 82.47 | 79.42 | 82.03 |
| | | **MosaiCLIP** | **90.18** | **85.22** | **80.48** | **83.86** |
| | YFCC-15M | CLIP | 82.61 | 76.21 | 72.27 | 75.60 |
| | | NegCLIP | 81.40 | 76.05 | 72.06 | 75.18 |
| | | **MosaiCLIP** | **84.25** | **79.83** | **73.29** | **77.42** |

Table 20: Detailed Pre-training results on the SVO-Probes dataset. See Sec. G.2 for more details.

| Benchmark → | | SVO-Probes | | | |
|---|---|---|---|---|---|
| | Method | Obj | Subj | Verb | Avg |
| | CLIP | 88.13 | 83.85 | 78.76 | 83.58 |
| CC-100K | CLIP-FT | 93.17 | 88.64 | 83.87 | 88.56 |
| | NegCLIP | 92.90 | 88.16 | **84.02** | 88.36 |
| | **MosaiCLIP**$_{\text{NoCurric}}$ | 93.37 | **89.74** | 83.62 | **88.91** |
| | **MosaiCLIP**$_{\text{WiSE-FT}}$ | 92.65 | 88.69 | 82.90 | 88.08 |
| | **MosaiCLIP** | **93.45** | 89.39 | 83.35 | 88.73 |
| YFCC-100K | CLIP-FT | 89.63 | 85.83 | **80.36** | 85.27 |
| | NegCLIP | 88.43 | 84.05 | 79.21 | 83.90 |
| | **MosaiCLIP**$_{\text{NoCurric}}$ | 89.49 | 85.59 | 79.83 | 84.97 |
| | **MosaiCLIP**$_{\text{WiSE-FT}}$ | 87.86 | 84.97 | 78.30 | 83.71 |
| | **MosaiCLIP** | **89.93** | **86.45** | 79.71 | **85.36** |
| COCO | CLIP-FT | 93.60 | 91.37 | 85.48 | 90.15 |
| | NegCLIP | 93.59 | 91.43 | **85.58** | 90.20 |
| | **MosaiCLIP**$_{\text{NoCurric}}$ | 94.14 | 92.22 | 84.23 | 90.20 |
| | **MosaiCLIP**$_{\text{WiSE-FT}}$ | 93.13 | 92.07 | 83.75 | 89.65 |
| | **MosaiCLIP** | **94.16** | **93.04** | 84.82 | **90.67** |

Table 19: Detailed Fine-tuning results on the SVO-Probes dataset. See Sec. G.2 for more details.

set to 0.1. Training is performed using 64 NVIDIA A100 GPUs. NegCLIP's hard negative text creation method often results in no negative text for some texts in the pre-training dataset. Removing all such image-text pairs with no possible hard negative text results in poor performance for NegCLIP (due to fewer data to pre-train on). If we include these image-text pairs, the text batch size might differ for different GPUs since some image-text pairs are without hard negative texts and this causes instabilities. We hence keep a cache of sentences from previous batches and add it to the batch as negative examples so that all GPUs have the same text batch size during training. The same is done for MosaiCLIP since not all images might have the same number of unique positive and negative sub-graphs available. For NegCLIP we create hard negative sentences using code released by (Yuksek-

| (Pre-training, Fine-tuning) data → | | CC-12M, CC-100K | | | | YFCC-15M, YFCC-100K | | | |
|---|---|---|---|---|---|---|---|---|---|
| Retrieval Set → | | Comp | | Atom | | Comp | | Atom | |
| Method | Split → | Seen | Unseen | Seen | Unseen | Seen | Unseen | Seen | Unseen |
| CLIP | | 48.3 | 45.1 | 39.2 | 35.0 | 42.0 | 39.8 | 43.4 | 39.5 |
| CLIP-FT | | 48.5 | 45.8 | 40.0 | 35.6 | 39.1 | 36.4 | 42.4 | 38.3 |
| NegCLIP | | 55.1 | 53.1 | 41.5 | 37.5 | 41.9 | 38.8 | 42.8 | 39.0 |
| MosaiCLIP$_{\text{NoCurric}}$ | | 71.4 | 71.1 | 45.3 | 40.2 | 50.1 | 46.6 | 44.9 | 40.5 |
| MosaiCLIP$_{\text{WiSE-FT}}$ | | 68.4 | 67.5 | 46.1 | **41.2** | 48.9 | 48.1 | **46.2** | **43.6** |
| **MosaiCLIP** | | **73.1** | **72.4** | **46.2** | 40.9 | **52.3** | 48.8 | 45.7 | 41.5 |

Table 21: Fine-tuning results on the CREPE - Systematicity datasets. We take OpenCLIP models pre-trained on CC-12M, and YFCC-15M, fine-tune them on CC-100K, and YFCC-100K, respectively, and test them on CC-12M, YFCC-15M split of CREPE dataset, respectively. See Sec. 4.1 for more details. We recalculate CLIP results since Ma et al. (2022) do not normalize CLIP embeddings before taking the dot product for text and image embeddings, resulting in an incorrect score.

| Arch. | Pre-training data → | | CC-12M | | | | YFCC-15M | | | |
|---|---|---|---|---|---|---|---|---|---|---|
| | Retrieval Set → | | Comp | | Atom | | Comp | | Atom | |
| | Method ↓ | Split → | Seen | Unseen | Seen | Unseen | Seen | Unseen | Seen | Unseen |
| Swin-T | CLIP | | 45.9 | 44.1 | 41.7 | 37.3 | 40.2 | 39.6 | 42.9 | 41.7 |
| | NegCLIP | | 76.4 | 80.3 | 45.1 | 39.6 | 47.3 | 47.1 | 43.2 | 41.5 |
| | **MosaiCLIP** | | **85.3** | **92.1** | **49.3** | **44.5** | **80.7** | **89.6** | **48.2** | **45.3** |
| RN-50 | CLIP | | 44.9 | 42.9 | 40.9 | 36.7 | 38.7 | 38.9 | 40.6 | 38.9 |
| | NegCLIP | | 78.6 | 82.0 | 46.8 | 41.4 | 61.5 | 67.2 | 43.5 | 41.5 |
| | **MosaiCLIP** | | **85.3** | **92.6** | **47.8** | **44.4** | **80.1** | **90.2** | **46.6** | **45.0** |

Table 22: Pre-training results on CREPE - Systematicity datasets. Models are pre-trained using CC-12M and YFCC-15M datasets and tested on the corresponding CC-12M and YFCC-15M split of the CREPE dataset. Results for both backbones Swin-Tiny and RN-50 are shown. See Sec. 4.1 for more details.

gonul et al., 2022). For MosaiCLIP training, for each image, we always use one hard negative text created using NegCLIP's swapping technique, followed by positive and negative subgraphs created using our method. Sub-graph sampling probabilities are kept as $p_2 = p_3$, $p_1 = 0.15$.

## H.2 Tree-Score details:

Murty et al. (2022) devised a method to calculate the tree-score of a transformer over a given dataset of sentences $\mathbb{D}$. This tree-score measures the functional tree-structuredness of a given transformer encoder. See Murty et al. (2022) for exact details for the algorithm to calculate the tree-scores. We use the code released by the authors[5] for the purpose of calculating tree-scores for CLIP's language encoder. In practice we use 5K sentences from the COCO-validation set as the held ouot test set $\mathbb{D}$ over which we calculate the tree-scores.

[5]https://github.com/MurtyShikhar/TreeProjections

## H.3 Computing Infrastructure and Run-Time:

We use NVIDIA A100 GPUs for all our experiments. Pre-training experiments took about 1.5 days per model while using 64 GPUs. Fine-tuning experiments on CC-FT, YFCC-FT and COCO took about 45 mins each and experiments on CC3M took 5 hours per model, while using 4 GPUs.

## H.4 Model Parameters:

We use standrad CLIP models and as part of all models, is a transformer language encoder having 12 layers, 8 attention heads and 512 as it's width. For vision encoders we use 1. ResNet-50 hvaing 23M trainable parameters and 2. Transformer vision encoders a) Swin-Tiny with patch-size 4 and window size 7 following (Yang et al., 2022) and b) ViT-B-32 which has patch size 32, 12 layers and 12 attention heads.

## H.5 Evaluation Metrics:

Strictly following the respective papers and released code[6], for ARO, VL-Checklist, SVO we use accuracy as the metric as defined by the respecitve papers. And for CREPE-Productivty, and CREPE-Systematicity [7] we use Recall@1 as our metric of evaluation.

## H.6 Summary Statistics of results:

We provide standard deviation results using 3 random seeds in Appendix Section D.2 for Fine-tuning experiments on the CC-FT dataset. For all other datasets, including the expensive pre-training runs we use a single seed for our experiments.

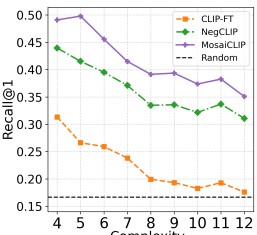
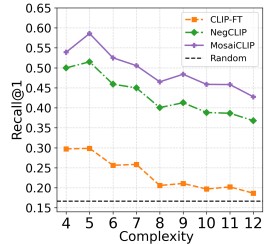

(a) Finetuning data: CC-100k

(b) Finetuning data: COCO

Figure 8: Fine-tuninig Results on CREPE - Productivity (generalization to longer and more complex sentences). Fine-tuning datasets are mentioned below each figure.

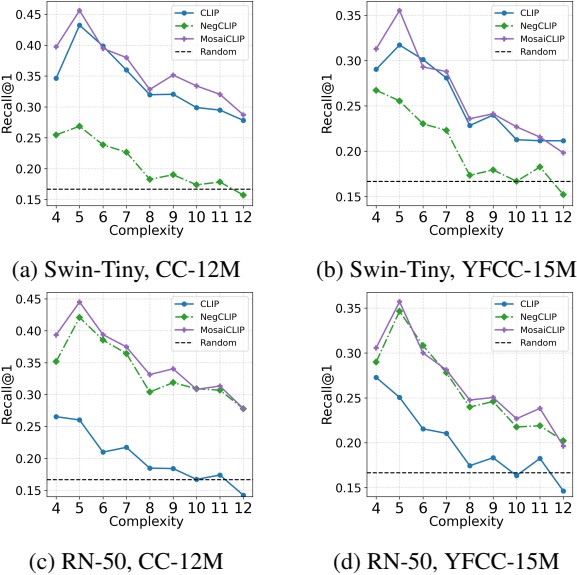

(a) Swin-Tiny, CC-12M

(b) Swin-Tiny, YFCC-15M

(c) RN-50, CC-12M

(d) RN-50, YFCC-15M

Figure 9: Pre-Training Results on CREPE - Productivity (generalization to longer and more complex sentences). Pre-Training model and datasets are mentioned below each figure.

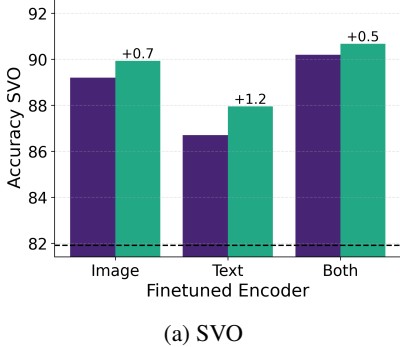

(a) SVO

Figure 10: Extension of Figure 4 c), d). Selectively fine-tuning of image, text encoders and measure performance on SVO-Probes dataset.

---

[6]ARO: https://github.com/mertyg/vision-language-models-are-bows, SVO-Probes https://github.com/deepmind/svo_probes, VL-Checklist: https://github.com/om-ai-lab/VL-CheckList

[7]CREPE Code: https://github.com/RAIVNLab/CREPE

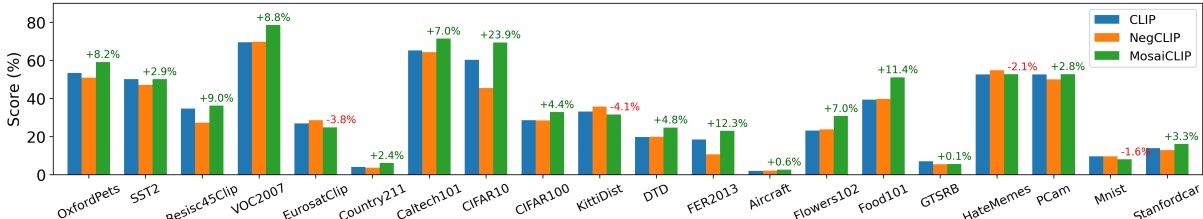

Figure 11: Comparing of CLIP, NegCLIP and MosaiCLIP on 20 datasets of from the ELEVATER (Li et al., 2022a) benchmark. Models in this graph are pretrained with CC-12M data and have Swin-Tiny as the vision backbone. See Sec. 4.1 for more details.

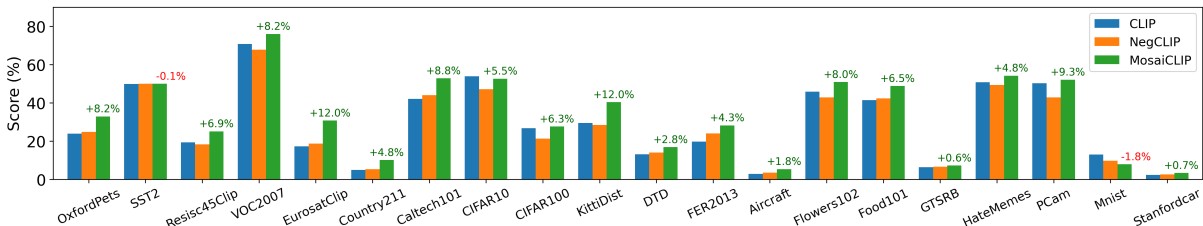

Figure 12: Comparing of CLIP, NegCLIP and MosaiCLIP on 20 datasets of from the ELEVATER (Li et al., 2022a) benchmark. Models in this graph are pretrained with YFCC-15M data and have Swin-Tiny as the vision backbone. See Sec. 4.1 for more details.

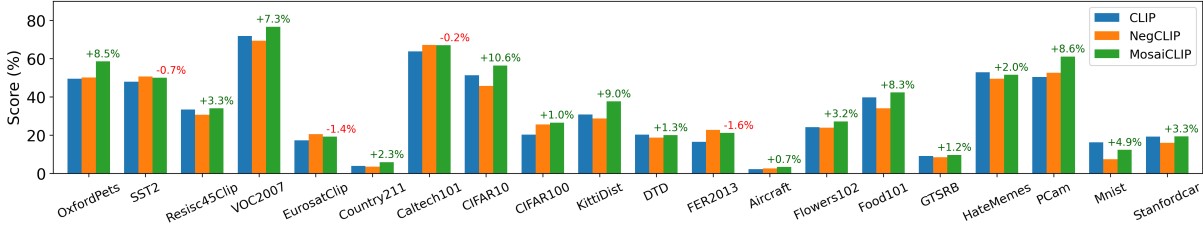

Figure 13: Comparing of CLIP, NegCLIP and MosaiCLIP on 20 datasets of from the ELEVATER (Li et al., 2022a) benchmark. Models in this graph are pretrained with CC-12M data and have ResNet-50 as the vision backbone. See Sec. 4.1 for more details.

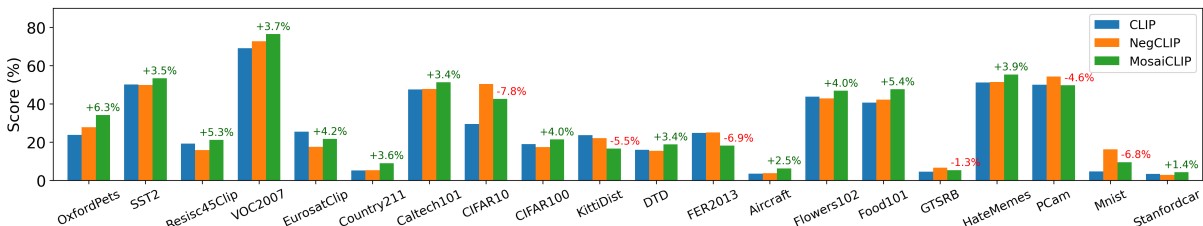

Figure 14: Comparing of CLIP, NegCLIP and MosaiCLIP on 20 datasets of from the ELEVATER (Li et al., 2022a) benchmark. Models in this graph are pretrained with YFCC-15M data and have ResNet-50 as the vision backbone. See Sec. 4.1 for more details.