# OpenReview forum: "Coarse-to-Fine Contrastive Learning in Image-Text-Graph Space for Improved Vision-Language Compositionality"
_EMNLP/2023/Conference — EMNLP 2023 Main_

### Official Review · Reviewer_oZZx · 2023-07-30

**Soundness:** 4

**Excitement:**

4: Strong: This paper deepens the understanding of some phenomenon or lowers the barriers to an existing research direction.

**Missing References:**

N/A

**Paper Topic And Main Contributions:**

This paper aims to improve the compositional reasoning ability of vision-language models (VLMs). To address this problem, a coarse-to-fine contrastive learning framework (MosaiCLIP) is proposed, which has three major technical innovations:
* **Scene graph guided text decomposition**: The image caption is converted into a graph structure using a text scene graph parser and the sub-graphs are extracted as positive samples in image-text contrastive learning. Specifically, the sub-graphs are converted to sentences based on a template before being fed to the VLM text encoder.
* **Negative sub-graph creation**: Hard negative sub-graphs are created by three operations, i.e., (1) node swapping and replacement, (2) edge replacement and (3) Connecting sub-graphs.
* **Curriculum and robust fine-tuning**: To bridge the gap in training objectives between conventional CLIP model and MosaiCLIP, a two-stage fine-tuning strategy is proposed. In the first stage, each image is only associated with a single positive sub-graph and a negative sub-graph. In the second stage, each image has multi-stage positive and negative sub-graphs.

The empirical studies demonstrate that MosaiCLIP outperforms existing CLIP-based VLMs in terms of visio-linguistic compositional reasoning. Moreover, MosaiCLIP is more robust when the image captions in the training data are noisy. The reason behind the improvement of MosaiCLIP is also explained based on the improved Tree-Score of the text encoder.

**Questions For The Authors:**

What are the templates when there are multiple nodes in the sub-graphs? Could you provide some specific examples?

**Reasons To Accept:**

* The proposed coarse-to-fine contrastive learning framework, which targets the viso-linguistic compositional reasoning problem, is intuitive and each component of MosaiCLIP is well-designed.
* The experiments are comprehensive and the results are convincing, which validate the superiority of MosaiCLIP in viso-linguistic compositional reasoning. The ablation studies suggest that every component of MosaiCLIP is conducive.
* The analysis on the reason behind MosaiCLIP’s improvement is instructive.
* The limitations of this work are properly recognized and discussed.
* The paper is well-written and easy to follow.

**Reasons To Reject:**

* As discussed in the Limitation section, it is unclear whether the proposed method can bring improvement to more advanced VLMs like BLIP.
* There is no evaluation on standard image-text retrieval tasks (e.g., on COCO), in addition to the evaluation on compositional reasoning benchmarks.

**Reproducibility:**

4: Could mostly reproduce the results, but there may be some variation because of sample variance or minor variations in their interpretation of the protocol or method.

**Reviewer Confidence:**

3: Pretty sure, but there's a chance I missed something. Although I have a good feel for this area in general, I did not carefully check the paper's details, e.g., the math, experimental design, or novelty.

**Typos Grammar Style And Presentation Improvements:**

Some abbreviations are inconsistent, e.g., Fig.3 (line 452) and Figures 4 (line 467).

---

> ### Author Rebuttal · Authors · 2023-08-29
>
> We thank the reviewer for reviewing our paper in great detail, for recognizing our technical innovations, and believing that MosaiCLIP is well-designed, with the results being comprehensive and convincing and each component of MosaiCLIP being important. We also thank the reviewer for recognizing our efforts in writing a sound paper with properly recognized limitations.
>
> Please see our response below:
>
>
> >#### **1. As discussed in the Limitation section, it is unclear whether the proposed method can bring improvement to more advanced VLMs like BLIP.**
>
> We understand the reviewers concern, however as mentioned in our paper and limitations section, our primary focus is on two tower encoder models like CLIP, since they enjoy various benefits like strong zero shot generalization capabilities, inference efficiency, wide applicability etc due to which there have been numerous works in literature that try to improve them, for example [1,2,3]. Having said that, we believe our methods are model agnostic and aimed at improving contrastive learning through our coarse-to-fine learning framework and negative mining techniques. We believe our methods can directly be used to improve other VLMs like BLIP, X-VLM since these models also use contrastive learning while training. We are performing an experiment on BLIP to test this hypothesis and would be happy to provide the results soon during the discussion phase. We have very limited GPU resources during this rebuttal period (compared to when we submitted our paper), hence it is taking more time than expected.
>
> References
>
> 1. Mu et. al, SLIP: Self-supervision meets Language-Image Pre-training.
>
> 2. Li et. al, Supervision Exists Everywhere: A Data Efficient Contrastive Language-Image Pre-training Paradigm.
>
> 3. Goel et. al, CyCLIP: Cyclic Contrastive Language-Image Pretraining
>
>
> >#### **2. There is no evaluation on standard image-text retrieval tasks (e.g., on COCO), in addition to the evaluation on compositional reasoning benchmarks.**
>
> Originally we followed previous works like Syn-CLIP and Teaching SVLC and used the ELEVATER and ImageNet benchmark as the datasets to show that improving compositional reasoning improves or maintains the representation learning capabilities of MosaiCLIP.
> We agree with the reviewer that it’s important to test our models on image-text retrieval tasks as well.
> See the table below for image-text retrieval scores in different settings, on COCO and Flickr30K datasets. We use the popular Karpathy splits and use the 5K and 1K test set for COCO and Flickr30k retrieval scores respectively:
>
>
> Models with Swin-Tiny backbone, pre-trained on YFCC-15M and tested on retrieval datasets:
>
> |           |   COCO      |            |    Flickr.    |           |    AVG.    |
> | -------------- | ------------ | ------------ | ------------ | ------------ | --------- |
> | Model     | I2T R@1 | T2I R@1 | I2T R@1 | T2I R@1 |      |
> | CLIP      | 20.7    | 13.1    | 36.2    | 24.1    | 23.5 |
> | NegCLIP   | 20.1    | 12.9    | 38.6    | 23.3    | 23.7 |
> | MosaiCLIP | **25.9**    | **16.5**    | **44.5**    | **29.5**    | **29.1** |
>
>
> OpenAI -CLIP Models with ViT-B-32 backbone, fine-tuned on COCO and tested on both retrieval datasets:
>
> |           | COCO    |         | Flickr  |         | AVG  |
> | --------- | ------- | ------- | ------- | ------- | ---- |
> | Model     | I2T R@1 | T2I R@1 | I2T R@1 | T2I R@1 |      |
> | CLIP      | 50.1    | 30.1    | 78.5    | 58.9    | 54.4 |
> | NegCLIP   | **56.2**    | 40.9    | **78.6**    | 67.3    | 60.8 |
> | MosaiCLIP | **56.1**    | **44.8**    | 75.6    | **70.2**    | **61.7** |
>
> **Observations:**
>
> a] During pre-training, MosaiCLIP significantly improves retrieval scores compared to NegCLIP and CLIP,, with an increase up to +6.2 Recall@1 points across T2I and I2T metrics.
>
> b] During fine-tuning, we see that performance of MosaiCLIP is better or equal to NegCLIP for COCO retrieval, while it’s worse than NegCLIP for text retrieval and better than NegCLIP for image retrieval. On average, MosaiCLIP improves over NegCLIP.
>
> c] Improvements in compositional reasoning improves or maintains performance on retrieval benchmarks.
>
>
> >#### **3. What are the templates when there are multiple nodes in the sub-graphs? Could you provide some specific examples?**
>
> Nodes can be either attributes or objects, here are some examples of templates with more than two nodes (Ni, Ai, Ri refers to objects, attributes and relations):
> For sub-graphs having 2 object nodes, where each may or may not have attribute nodes
> - {A1} {N1} {R1} {A2} {N2}
> - {N1} {R1} {A2} {N2}
> - {A1} {N1} {R1} {N2}
> - {N1} {R1} {N2}
>
> Example: “{Blue} {pen} {on top of} {white} {table}” will be a sentence as a result of the first template above
>
> For sub-graphs having 3 object nodes, where each may or may not have attribute nodes. We join two sentences and one.
> - {A1} {N1} {R1} {A2} {N2} and {A2} {N2} {R2} {A3} {N3}
> - {N1} {R1} {A2} {N2} and {A2} {N2} {R2} {A3} {N3}
> - {A1} {N1} {R1} {N2} and {N2} {R2} {A3} {N3}
> - {A1} {N1} {R1} {A2} {N2} and {A2} {N2} {R2} {N3}
> - … (variations where multiple objects dont have attributes)
>
> Example: Given a sub-graph having relations: (pen, on top of, table); (table, behind, bed) and objects pen, table, bed having attributes blue, white, comfortable -- we can have a sentence using the first template as:
> “{Blue} {pen} {on top of} {white} {table} and {white} {table} {behind} {comfortable} {bed}”
>
>
>
> >#### **4. Some abbreviations are inconsistent, e.g., Fig.3 (line 452) and Figures 4 (line 467).**
>
> Thank you for pointing this out. We will fix these in the final version of the paper.
>
>
> **Please let us know if you have any other questions or concerns and we would be happy to answer them and engage in discussions.**

---

### Official Review · Reviewer_JAaq · 2023-08-04

**Soundness:** 4

**Excitement:**

4: Strong: This paper deepens the understanding of some phenomenon or lowers the barriers to an existing research direction.

**Paper Topic And Main Contributions:**

This article presents a scene graph-based image-text contrastive learning method. By incorporating scene graphs, the fine-grained control of contrastive learning is achieved, and experimental results demonstrate performance improvement compared to the baseline.

**Reasons To Accept:**

1. It is commendable that the experiments in this study were conducted in a thorough and reliable manner, providing substantial evidence for the model's performance. The validation of the motivation behind the proposed approach adds further credibility to the research findings.

2. The method is indeed novel and inspiring, offering fresh perspectives in the field.

**Reasons To Reject:**

1. Indeed, the method's success heavily relies on the quality of scene graph generation. If errors occur during scene graph generation, it may lead to subsequent inaccuracies in the results. Ensuring a reliable and accurate scene graph generation process is crucial for the overall effectiveness of the approach.

2.The process of extracting scene graphs may consume significant computational resources, and in situations where the scene is complex, it might not be possible to obtain complete or accurate scene graph information. This can potentially harm the model's performance.

3.Compared to other state-of-the-art models in the same field, the performance of this method is not particularly outstanding.

**Reproducibility:**

4: Could mostly reproduce the results, but there may be some variation because of sample variance or minor variations in their interpretation of the protocol or method.

**Reviewer Confidence:**

3: Pretty sure, but there's a chance I missed something. Although I have a good feel for this area in general, I did not carefully check the paper's details, e.g., the math, experimental design, or novelty.

---

> ### Author Rebuttal · Authors · 2023-08-29
>
> We thank the reviewer for an in depth review of our paper, for recognizing our efforts in terms of thorough and reliable experimentation, and for believing that our work is novel, inspiring and brings fresh perspectives to the field.
> Please see our response below:
>
> >#### **1. Indeed, the method's success heavily relies on the quality of scene graph generation. If errors occur during scene graph generation, it may lead to subsequent inaccuracies in the results. Ensuring a reliable and accurate scene graph generation process is crucial for the overall effectiveness of the approach.**
>
> a] In our experiments, we make use of a simple and popular, rule based text scene graph parser (Wu et al., 2019) which is frequently used in literature and is based on the stanford scene graph parser. This parser is not the best parser available, and can contain some errors, such as missing edges or objects.
> Here are few examples sentences from the COCO Validation set, for which the corresponding scene graphs have some kind of error:
>
> | Sentence (Image Caption)                                                      | Error in scene graph                                         |
> | ------------------------------------------------------- | ----------------------------------------------------------------- |
> | A person driving a car on the street with a white car in view  | Missed a relation between car and street: (car, on, street)   |
> | Two men standing behind a cow in front of a house. | Missed relation between men and house: (men, in front of, house) |
> | A bathroom with a toilet and sink and a bathtub sitting on a hardfloor | Missed object: hardfloor
>
> We generally find most of the errors are in missed relations and few in missing objects.
>
> However, even these graphs with errors are useful because they contain informative sub-graphs that the MosaiCLIP method can use and perform coarse-to-fine learning. MosaiCLIP is tolerant to some type of errors, for example missing edges and objects, because even if some objects or edges are missing in the text scene graph, we can still create useful and even accurate positive and negative sub-graphs from the remaining graph and our formulation of image-text-graph alignment would still hold (as described in Section 3.2 of the paper). It is to be noted that the text scene graph is only used for constructing more positive/negative examples during training and it's **not used for inference** and hence it does not lead to any cascading inference time errors.
>
> b] To give more support, we refer to Analysis Section 4.2 of the paper, where we discuss that MosaiCLIP is robust to noise in data. A pre-training dataset like YFCC is very noisy as compared to CC-12M, and this noise leads to a less than optimal scene graph. However, meaningful sub-graphs are still present which lead to effective learning. We can imagine a similar effect will take place if the scene graph parser itself causes more errors and we expect that MosaiCLIP will be able to handle it to some extent.
>
> We do not search for SOTA scene graph parsers in our work, and find that our approach can already achieve good performance using a readily available open source parser. Replacing this parser with any better parser is straightforward and should result in better performance of our model.
>
>
> >#### **2. The process of extracting scene graphs may consume significant computational resources, and in situations where the scene is complex, it might not be possible to obtain complete or accurate scene graph information. This can potentially harm the model's performance.**
>
> We use a text scene graph parser and we believe the computational resources it consumes are significantly lesser as compared to the large GPU resources required for CLIP training, which makes the overhead marginal. We make the following points:
>
> a] Scene graph parsing happens on CPUs since it's largely a rule based system. We can perform multiprocessing to parallelize it on one machine, and we can use multiple machines in parallel as well, which is a common setup for practitioners training CLIP. No expensive GPU resources are used in this process.
>
> b] The Scene graph generation process can be done offline before training the model, and has to be done only one time for any number of experiments with a dataset. This will save computation required to parse scene graphs for the same data point on every epoch (32 epochs in the common pre-training setup).
>
> c] In our python implementation, YFCC-15M  can be parsed in 17.6 minutes on a single machine with 96vCPUs. This time is further reduced to under 5 minutes if 4-8 nodes are used. Compare this with the time taken for pre-training where we use 8 nodes having 8 GPUs each, i.e., 64 GPUs in total and train for about 1.5 days. Hence the cost of creating scene graphs is marginal as compared to total resources spent on training CLIP.
>
> We discussed the accuracy and noise of scene graph generation as part of the previous answer.
>
> >#### **3. Compared to other state-of-the-art models in the same field, the performance of this method is not particularly outstanding.**
>
> We would like to seek further clarification regarding the term "state-of-the-art models" as mentioned by the reviewer in the context of our work. We have endeavored to delve deeper into the reviewer’s question in our response below:
>
> a] Our primary focus in this paper is on improving vision-language compositional reasoning and in this process we want to maintain or improve the general representation learning performance of VLMs which make them useful for general tasks like classification and retrieval.
> We identified all SOTA baselines for the compositional reasoning tasks that we focus on, which include models like SynCLIP, Teaching-SVLC, NegCLIP and we compare them with MosaiCLIP in Tables 1,2,3.
>
> In the broader field of Vision-Language representation learning, there exists other SOTA baselines like BLIP, XVLM, Flava which show good classification and retrieval performance on datasets like ImageNet and MSCOCO, however, they lack severely in terms of compositional reasoning as shown by [1,2]. Hence these methods can’t be considered SOTA for tasks related to compositional reasoning like attribute binding, relational reasoning, order understanding.
>
> If by sota models, the reviewer is referring to general VL models like BLIP, XVLM, Flava, we would be happy to include comparisons with these models. We compare them in the table below:
>
> |           | Rel  | Attr | COCO-Order | FLickr-Order | Average |
> | --------- | ---- | ---- | ---------- | ------------ | ------- |
> | Random    | 50   | 50   | 20         | 20           | 35    |
> | BLIP      | 59   | 88   | 32.1       | 36.9         | 54.0    |
> | XVLM      | 73   | **87**   | 36.2       | 46.3         | 60.6    |
> | FLAVA     | 25   | 73   | 3.9        | 12.9         | 28.7    |
> | CLIP      | 59.8 | 63.2 | 59.2       | 47.5         | 57.4    |
> | MosaiCLIP | **82.6** | 78.0 | **86.3**       | **87.9**         | **83.7**   |
>
> Accuracies for BLIP, XVLM, Flava are taken from [1].
>
> We observe that MosaiCLIP has better overall compositional reasoning capability as compared to baselines considered above which shows that it is possible to have the benefits of CLIP (eg. inference efficiency, strong zero-shot generalization capabilities) while also having improved compositionality. In particular, MosaiCLIP is better at relational reasoning and word order understanding, while also closing the gap between CLIP and the best model (XVLM) in terms of attribute binding.
>
> We will include this table in the final version of the paper for the benefit of future readers.
>
> b] We would also like to mention that our methods are model agnostic and aimed at improving contrastive learning through our coarse-to-fine learning framework and negative mining techniques. We believe our methods can directly be used to improve other VLMs like BLIP, X-VLM since these models also use contrastive learning while training. We are performing an experiment on BLIP to test this hypothesis and would be happy to provide the results soon during the discussion phase. We have very limited GPU resources during this rebuttal period (compared to when we submitted our paper), hence it is taking more time than expected.
>
> Please let us know if your question/concern has not been answered through our response, we would be happy to engage in more detailed discussion.
>
> References
>
> 1. Yuksekgonul et al, When and why vision-language models behave like bag-of-words models, and what to do about it? ICLR 2023
> 2. Zhao et. al, VL-CheckList: Evaluating Pre-trained Vision-Language Models with Objects, Attributes and Relations. EMNLP 2022
>
>
> **Please let us know if you have any other questions or concerns and we would be happy to answer them and engage in discussions.**

---

### Official Review · Reviewer_Ut7T · 2023-08-07

**Soundness:** 3

**Excitement:**

3: Ambivalent: It has merits (e.g., it reports state-of-the-art results, the idea is nice), but there are key weaknesses (e.g., it describes incremental work), and it can significantly benefit from another round of revision. However, I won't object to accepting it if my co-reviewers champion it.

**Paper Topic And Main Contributions:**

The paper proposed MosaiCLIP, a framework to decompose text into scene graphs for image-text contrastive learning. It incorporates hard-negative mining via text scene graph transformations and provides a coarse-to-fine contrastive learning strategy. The efficacy of MosaiCLIP is validated through comprehensive experiments across multiple architectures, datasets, training fashions, and compositional benchmarks.

**Reasons To Accept:**

1. They proposed to parse scene graphs from the text for contrastive Image-text pre-training. The text scene graphs enable multiple positive samples and hard negative mining, which facilitate contrastive training. The idea is interesting and novel to some degree.
2. The experimental result is impressive, showing a decent gain of the proposed model over previous methods.

**Reasons To Reject:**

1. Lack of enough comparison with previous works. There are also other works utilizing more types of negative samples such as DeCLIP, etc, which is not compared in the experiments.
2. I wonder if the performance improvement is brought by the proposed method or just a larger batch size brought by more negative samples. Are NegCLIP/CLIP and the proposed method in comparison using the same text batch size? If not so, it's a necessary comparison that adds more negative text samples in the original CLIP or NegCLIP so that the total text bz is the same as the proposed method.
3. Evaluation on image-text retrieval is missed.

**Reproducibility:**

4: Could mostly reproduce the results, but there may be some variation because of sample variance or minor variations in their interpretation of the protocol or method.

**Reviewer Confidence:**

4: Quite sure. I tried to check the important points carefully. It's unlikely, though conceivable, that I missed something that should affect my ratings.

---

> ### Author Rebuttal · Authors · 2023-08-29
>
> We thank the reviewer for an in depth review of our paper, for believing that our idea is interesting and novel, and for considering the experimental results as impressive. We also thank the reviewer for suggesting important experiments which we believe would greatly enhance the paper for future readers.
>
> Please see our response below:
>
> >#### **1. Lack of enough comparison with previous works. There are also other works utilizing more types of negative samples such as DeCLIP, etc, which is not compared in the experiments.**
>
> **Difference w.r.t methods like DeCLIP:** Our negative mining strategy focuses on improving compositional reasoning capabilities of Vision Language Models and hence we compare MosaiCLIP with all previous methods that improve vision-language compositional reasoning (SynCLIP, Teaching-SVLC, NegCLIP). Our focus on compositionality is different and orthogonal to methods like DeCLIP that do not use negative samples focused on compositional reasoning capabilities like relation understanding or attribute binding and also do not perform coarse-to-fine learning like in our case.
>
> DeCLIP uses extra losses and negative mining strategies, including self-supervision on images and text, multi-view supervision and nearest neighbor supervision to improve their method’s data efficiency. Most negative mining techniques studied in literature do-not promote compositional reasoning. We find that although these additional losses provide extra supervision for data-efficient learning, the problem of poor compositional reasoning still remains (as shown below).
>
> See the following table for results of DeCLIP and a related method, SLIP on attribute, relation, order understanding and systematic generalization. Here, we consider models pretrained on YFCC-15M obtained from DeCLIP github repository and compare it with MosaiCLIP pretrained on YFCC-15M. (best results of RN-50 architecture models are in **bold**, best VIT-B-32 architecture results in **_bold+italics_**)
>
> |           |        | ARO  |      |            |              | CREPE     |             |           |             |
> | --------- | ------ | ---- | ---- | ---------- | ------------ | --------- | ----------- | --------- | ----------- |
> |           | Arch   | Rel  | Attr | COCO-Order | FLickr-Order | Comp-Seen | Comp-Unseen | Atom-Seen | Atom-Unseen |
> | Random    | \-     | 50   | 50   | 20         | 20           | 14.3      | 14.3        | 20        | 20          |
> | DeCLIP    | ViTB32 | 55.3 | 57.7 | 19.3       | 20.6         | 40.3      | 43.7        | 45.6      | _**46.6**_        |
> | SLIP      | ViTB32 | 59.6 | 57.0 | 23.7       | 25.0         | 44.7      | 44.1        | 45.4      | 43.8        |
> | CLIP      | RN-50  | 57.8 | 55.1 | 18.5       | 18.0         | 38.7      | 38.9        | 40.6      | 38.9        |
> | DeCLIP    | RN-50  | 54.0 | 56.3 | 17.4       | 18.1         | 38.8      | 41.1        | 45.8      | **44.8**        |
> | NegCLIP   | RN-50  | 68.0 | 58.5 | 36.9       | 37.3         | 61.5      | 67.2        | 43.5      | 41.5        |
> | MosaiCLIP | RN-50  | **76.3** | **68.9** | **37.6**       | **38.9**         | **80.1**      | **90.2**        | **46.6**      | **45.0**        |
>
>
> **Observations:**
>
> a] DeCLIP is slightly worse than CLIP on attribute, relation and order understanding, and better than CLIP on systematic generalization capabilities as measured by the CREPE dataset
>
> b] NegCLIP is a stronger compositional reasoner as compared to DeCLIP, SLIP in general as depicted by most of the above results.
>
> c] MosaiCLIP improves over DeCLIP and SLIP in terms of compositional reasoning.
>
> Our method is model agnostic and orthogonal to DeCLIP, hence the extra losses/self-supervision used by DeCLIP can also be used in our framework to further improve data efficiency and performance of our approach. This is left as future work but we would work on implementing this and hope to include this experiment in the final version of our paper.
>
> >#### **2. I wonder if the performance improvement is brought by the proposed method or just a larger batch size brought by more negative samples. Are NegCLIP/CLIP and the proposed method in comparison using the same text batch size? If not so, it's a necessary comparison that adds more negative text samples in the original CLIP or NegCLIP so that the total text bz is the same as the proposed method.**
>
>
> Thank you for bringing this up, this is an important consideration to establish the usefulness of our method. We were aware of this and have provided experiments to study this in the paper itself.  **See Appendix A.3.6** where we provide an analysis of the computational cost of our Model vs NegCLIP. **To summarize and answer your questions**,
>
> a] MosaiCLIP’s effective batch size is larger compared to CLIP (by 6 to 9 times), due to text decomposition. NegCLIP’s effective batch size is larger by 8x since it uses image and text negative mining.
>
> b] MosaiCLIP uses a slightly larger effective batch size compared to NegCLIP in most of our experiments, (by a factor of 9/8=1.125). We use 3 positive and 6 negative sub-graphs per image (see Appendix section A.3.1, Fig 6 and 7 for more details on this choice).
>
> c] We find that providing NegCLIP with even larger batch sizes does not improve its performance, and performance stays below MosaiCLIP always. We also see a decrease in performance on some datasets when batch size for NegCLIP is increased which is possible because as batch size increases, negative images and texts sampled by NegCLIP can be false negatives for other elements of the batch.
>
> d] MosaiCLIP outperforms NegCLIP even with a lower batch size as shown in the table below.
>
>
> Here, a global batch size refers to batch size without any negative mining or text decomposition. Effective batch size refers to C=n*m where n and m are the number of images and text after negative mining/decomposition.  B=256 In the following table.
>
>
>
> | Method    | Global batch size (#I \* #T) | Effective Batch size, C=n\*m | ARO  |       | | SVO   |
> | --------- | ----------------------------- | ----------------------------------------------------- | ---- | ----- |  ----- | ----- |
> |           |                               |                                                       | Rel. | Attr. | Ord. | Avg.  |
> | MosaiCLIP | B\*B                          | B\*9B = 9B\*B                                         | **79.0** | **69.6**  | **85.5** | **88.91** |
> |           | B\*B                          | B\*6B = 6B\*B                                         | 78.2 | 68.0  | 85.3| **88.92** |
> | NegCLIP   | B\*B                          | 2B\*4B = 8B\*B                                        | 71.5 | 65.4  | 84.5 | 88.36 |
> |           | 2B\*2B                        | 4B\*8B = 32B\*B                                       | 68.9 | 65.6  | 83.9| 88.68 |
> |           | 4B\*4B                        | 8B\*16B = 128B\*B                                     | 67.6 | 65.1  | 84.3| **88.93** |
> |           | 8B\*8B                        | 16B\*32B = 512B\*B                                    | 65.7 | 64.2  | 83.8| 88.72 |
> |           | 16B\*16B                      | 32B\*64B = 2048B\*B                                   | 62.5 | 63.7  | 82.4| 88.11 |
>
>
>
>
> Along with this, in the paper we provide additional analysis showing that MosaiCLIP can improve over NegCLIP with as low as 0.3x-0.6x of the total data, which can further reduce the computational cost. Computational cost can also be traded off with performance by reducing or increasing the number of positive and negative subgraphs sampled as shown in Table 6 and 7 of the paper.
>
>
> **Providing CLIP with a larger batch size:**
>
> [1,2] measure various OpenAI-CLIP and OpenCLIP models trained on up to 32k batch size and find that the compositional reasoning performance is still very poor. An intuition for this finding is that, even if the batch size scales to very large values, it is very unlikely that close hard negatives for sentences and images exist in the same batch, which are essential for the model to learn to distinguish between changes in relations or attributes of objects. Due to this even with very large batch size, the required fine-grained understanding skills are not learned.
>
>
> References
>
> 1. Yuksekgonul et al, When and why vision-language models behave like bag-of-words models, and what to do about it? ICLR 2023
>
> 2. Ma et al, CREPE: Can Vision-Language Foundation Models Reason Compositionally? CVPR 2023
>
>
> >#### **3. Evaluation on image-text retrieval is missed.**
>
> Originally we followed previous works like Syn-CLIP and Teaching SVLC and used the ELEVATER and ImageNet benchmark as the datasets to show that improving compositional reasoning improves or maintains the representation learning capabilities of MosaiCLIP.
> We agree with the reviewer that it’s important to test our models on image-text retrieval tasks as well.
> See the table below for image-text retrieval scores in different settings and different datasets including COCO and Flickr30K. We use the popular Karpathy splits and use the 5K and 1K test set for COCO and Flickr30k retrieval scores respectively:
>
>
> Models with Swin-Tiny backbone, pre-trained on YFCC-15M and tested on retrieval datasets:
>
> |           |   COCO      |            |    Flickr.    |           |    AVG.    |
> | -------------- | ------------ | ------------ | ------------ | ------------ | --------- |
> | Model     | I2T R@1 | T2I R@1 | I2T R@1 | T2I R@1 |      |
> | CLIP      | 20.7    | 13.1    | 36.2    | 24.1    | 23.5 |
> | NegCLIP   | 20.1    | 12.9    | 38.6    | 23.3    | 23.7 |
> | MosaiCLIP | **25.9**    | **16.5**    | **44.5**    | **29.5**    | **29.1** |
>
>
> OpenAI -CLIP Models with ViT-B-32 backbone, fine-tuned on COCO and tested on both retrieval datasets:
>
> |           | COCO    |         | Flickr  |         | AVG  |
> | --------- | ------- | ------- | ------- | ------- | ---- |
> | Model     | I2T R@1 | T2I R@1 | I2T R@1 | T2I R@1 |      |
> | CLIP      | 50.1    | 30.1    | 78.5    | 58.9    | 54.4 |
> | NegCLIP   | **56.2**    | 40.9    | **78.6**    | 67.3    | 60.8 |
> | MosaiCLIP | **56.1**    | **44.8**    | 75.6    | **70.2**    | **61.7** |
>
>
> **Observations:**
>
> a] During pre-training, MosaiCLIP significantly improves retrieval scores compared to NegCLIP and CLIP,, with an increase up to +6.2 Recall@1 points across T2I and I2T metrics.
>
> b] During fine-tuning, we see that performance of MosaiCLIP is better or equal to NegCLIP for COCO retrieval, while it’s worse than NegCLIP for text retrieval and better than NegCLIP for image retrieval. On an average, MosaiCLIP improves over NegCLIP.
>
> **Please let us know if you have any other questions or concerns and we would be happy to answer them and engage in discussions.**

---

### Meta-Review · Area_Chair_jjB7 · 2023-09-09

**Recommendation:** 5

**Metareview:**

The paper tries to improve the concept compositionally for contrastive VL models. The main novelty comes from the reliable negative example constructions from text scene graph. Overall, it is a solid paper.

---

### Decision · Program_Chairs · 2023-10-07

**Decision:**

Accept-Main

**Comment:**

The paper tries to improve the concept compositionally for contrastive VL models. The main novelty comes from the reliable negative example constructions from text scene graph. Overall, it is a solid paper.